# DiffuseVAE: Efficient, Controllable and High-Fidelity Generation from Low-Dimensional Latents

**Kushagra Pandey**                                                            *pandeyk1@uci.edu*
*Department of Computer Science*
*University of California, Irvine*

**Avideep Mukherjee**                                                  *avideep@cse.iitk.ac.in*
*Department of Computer Science*
*Indian Institute of Technology, Kanpur*

**Piyush Rai**                                                            *piyush@cse.iitk.ac.in*
*Department of Computer Science*
*Indian Institute of Technology, Kanpur*

**Abhishek Kumar**                                                    *abhishk@google.com*
*Google Research, Brain Team*

**Reviewed on OpenReview:** *https://openreview.net/forum?id=ygoNPRiLxw*

## Abstract

Diffusion probabilistic models have been shown to generate state-of-the-art results on several competitive image synthesis benchmarks but lack a low-dimensional, interpretable latent space, and are slow at generation. On the other hand, standard Variational Autoencoders (VAEs) typically have access to a low-dimensional latent space but exhibit poor sample quality. We present DiffuseVAE, a novel generative framework that integrates VAE within a diffusion model framework, and leverage this to design novel conditional parameterizations for diffusion models. We show that the resulting model equips diffusion models with a low-dimensional VAE inferred latent code which can be used for downstream tasks like controllable synthesis. The proposed method also improves upon the speed vs quality trade-off exhibited in standard unconditional DDPM/DDIM models (for instance, **FID of 16.47 vs 34.36** using a standard DDIM on the CelebA-HQ-128 benchmark using **T=10** reverse process steps) without having explicitly trained for such an objective. Furthermore, the proposed model exhibits synthesis quality comparable to state-of-the-art models on standard image synthesis benchmarks like CIFAR-10 and CelebA-64 while outperforming most existing VAE-based methods. Lastly, we show that the proposed method exhibits inherent generalization to different types of noise in the conditioning signal. For reproducibility, our source code is publicly available at `https://github.com/kpandey008/DiffuseVAE`.

## 1 Introduction

Generative modeling is the task of capturing the underlying data distribution and learning to generate novel samples from a posited explicit/implicit distribution of the data in an unsupervised manner. Variational Autoencoders (VAEs) (Kingma & Welling, 2014; Rezende & Mohamed, 2016) are a type of explicit-likelihood based generative models which are often also used to learn a low-dimensional latent representation for the data. The resulting framework is very flexible and can be used for downstream applications, such as learning disentangled representations (Higgins et al., 2017; Chen et al., 2019; Burgess et al., 2018), semi-supervised learning (Kingma et al., 2014), anomaly detection (Pol et al., 2020), among others. However, in image synthesis applications, VAE generated samples (or reconstructions) are usually blurry and fail to incorporate

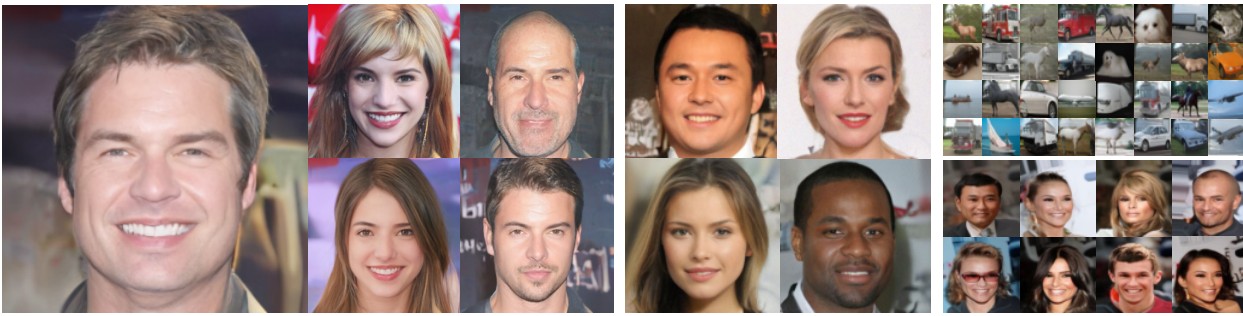

Figure 1: DiffuseVAE generated samples on the CelebA-HQ-256 (Left), CelebA-HQ-128 (Middle), CIFAR-10 (Right, Top) and CelebA-64 (Right, Bottom) datasets using just **25**, **10**, **25** and **25** time-steps in the reverse process for the respective datasets. The generation is entirely driven by low dimensional latents – the diffusion process latents are fixed and shared between samples after the model is trained (See Section 4.2 for more details).

high-frequency information (Dosovitskiy & Brox, 2016). Despite recent advances (van den Oord et al., 2018; Razavi et al., 2019; Vahdat & Kautz, 2021; Child, 2021; Xiao et al., 2021) in improving VAE sample quality, most VAE-based methods require large latent code hierarchies. Even then, there is still a significant gap in sample quality between VAEs and their implicit-likelihood counterparts like GANs (Goodfellow et al., 2014; Karras et al., 2018; 2019; 2020b).

In contrast, Diffusion Probabilistic Models (DDPM) (Sohl-Dickstein et al., 2015; Ho et al., 2020) have been shown to achieve impressive performance on several image synthesis benchmarks, even surpassing GANs on several such benchmarks (Dhariwal & Nichol, 2021; Ho et al., 2021). However, conventional diffusion models require an expensive iterative sampling procedure and lack a low-dimensional latent representation, limiting these models' practical applicability for downstream applications.

We present DiffuseVAE, a novel framework which combines the best of both VAEs and DDPMs in an attempt to alleviate the aforementioned issues with both types of model families. We present a novel two-stage conditioning framework where, in the first stage, any arbitrary conditioning signal ($y$) can be first modeled using a standard VAE. In the second stage, we can then model the training data ($x$) using a DDPM conditioned on $y$ and the low-dimensional VAE latent code representation of $y$. With some simplifying design choices, our framework reduces to a *generator-refiner* framework which involves fitting a VAE on the training data ($x$) itself in the first stage followed by modeling $x$ in the second stage using a DDPM conditioned on the VAE reconstructions ($\hat{x}$) of the training data,. The main contributions of our work can be summarized as follows:

1. **A novel conditioning framework**: We propose a generic DiffuseVAE conditioning framework and show that our framework can be reduced to a simple *generator-refiner* framework in which blurry samples generated from a VAE are *refined* using a conditional DDPM formulation (See Fig.2). This effectively equips the diffusion process with a low dimensional latent space. As a part of our conditioning framework, we explore two types of conditioning formulations in the second stage DDPM model.

2. **Controllable synthesis from a low-dimensional latent**: We show that, as part of our model design, major structure in the DiffuseVAE generated samples can be controlled directly using the low-dimensional VAE latent space while the diffusion process noise controls minor stochastic details in the final generated samples.

3. **Better speed vs quality tradeoff**: We show that DiffuseVAE inherently provides a better speed vs quality tradeoff as compared to a standard DDPM model on several image benchmarks. Moreover, combined with DDIM sampling (Song et al., 2021a), the proposed model can generate plausible samples in as less as 10 reverse process sampling steps (For example, the proposed method achieves

an FID (Heusel et al., 2018) of 16.47 as compared to 34.36 by the corresponding DDIM model at T=10 steps on the CelebA-HQ-128 benchmark (Karras et al., 2018)).

4. **State of the art comparisons**: We show that DiffuseVAE exhibits synthesis quality comparable to recent state-of-the-art on standard image synthesis benchmarks like CIFAR-10 (Krizhevsky, 2009), CelebA-64 (Liu et al., 2015)) and CelebA-HQ (Karras et al., 2018) while maintaining access to a low-dimensional latent code representation.

5. **Generalization to different noises in the conditioning signal**: We show that a pre-trained DiffuseVAE model exhibits generalization to different noise types in the DDPM conditioning signal exhibiting the effectiveness of our conditioning framework.

## 2 Background

### 2.1 Variational Autoencoders

VAEs (Kingma & Welling, 2014; Rezende & Mohamed, 2016) are based on a simple but principled encoder-decoder based formulation. Given data $x$ with a latent representation $z$, learning the VAE is done by maximizing the evidence lower bound (ELBO) on the data log-likelihood, $\log p(x)$ (which is intractable to compute in general). The VAE optimization objective can be stated as follows

$$\mathcal{L}(\theta, \phi) = \mathbb{E}_{q_\phi(z|x)}[\log p_\theta(x|z)] - \mathcal{D}_{KL}[q_\phi(z|x)\|p(z)] \tag{1}$$

Under amortized variational inference, the approximate posterior on the latents, i.e., $(q_\phi(z|x))$, and the likelihood $(p_\theta(x|z))$ distribution can be modeled using deep neural networks with parameters $\phi$ and $\theta$, respectively, using the reparameterization trick (Kingma & Welling, 2014; Rezende & Mohamed, 2016). The choice of the prior distribution $p(z)$ is flexible and can vary from a standard Gaussian (Kingma & Welling, 2014) to more expressive priors (van den Berg et al., 2019; Grathwohl et al., 2018; Kingma et al., 2017).

### 2.2 Denoising Diffusion Probabilistic Models

DDPMs (Sohl-Dickstein et al., 2015; Ho et al., 2020) are latent-variable models consisting of a forward noising process $(q(x_{1:T}|x_0))$ which gradually destroys the structure of the data $x_0$ and a reverse denoising process $((p(x_{0:T})))$ which learns to recover the original data $x_0$ from the noisy input. The forward noising process is modeled using a first-order Markov chain with Gaussian transitions and is fixed throughout training, and the noise schedules $\beta_1$ to $\beta_T$ can be fixed or learned. The form of the forward process can be summarized as follows:

$$q(x_{1:T}|x_0) = \prod_{t=1}^{T} q(x_t|x_{t-1}) \tag{2}$$

$$q(x_t|x_{t-1}) = \mathcal{N}(\sqrt{1-\beta_t}x_{t-1}, \beta_t I) \tag{3}$$

$$q(x_t|x_0) = \mathcal{N}(\sqrt{\bar{\alpha}_t}x_0, (1-\bar{\alpha}_t)I) \text{ where } \alpha_t = (1-\beta_t) \text{ and } \bar{\alpha}_t = \prod_t \alpha_t \tag{4}$$

The reverse process can also be parameterized using a first-order Markov chain with a learned Gaussian transition distribution as follows

$$p(x_{0:T}) = p(x_T) \prod_{t=1}^{T} p_\theta(x_{t-1}|x_t) \tag{5}$$

$$p_\theta(x_{t-1}|x_t) = \mathcal{N}(\mu_\theta(x_t, t), \Sigma_\theta(x_t, t)) \tag{6}$$

Given a large enough $T$ and a well-behaved variance schedule of $\beta_t$, the distribution $q(x_T|x_0)$ will approximate an isotropic Gaussian. The entire probabilistic system can be trained end-to-end using variational inference. During sampling, a new sample can be generated from the underlying data distribution by sampling a latent

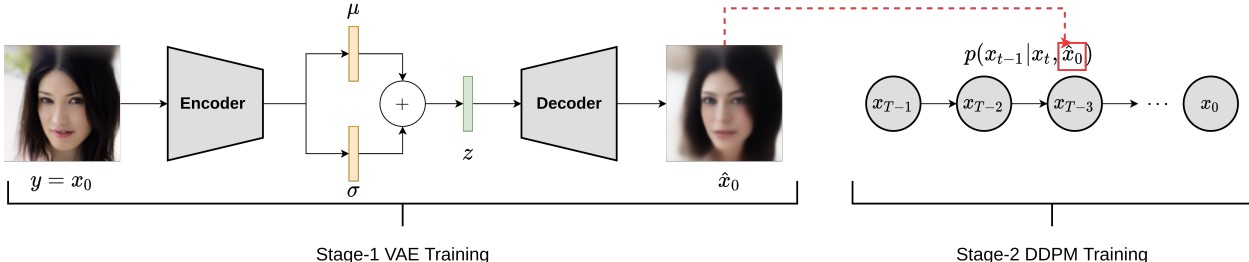

Figure 2: Proposed DiffuseVAE generative process under the simplifying design choices discussed in Section 3.2. DiffuseVAE is trained in a two-stage manner: The VAE encoder takes the original image $x_0$ as input and generates a reconstruction $\hat{x}_0$ which is used to condition the second stage DDPM.

(of the same size as the training data point $x_0$) from $p(x_T)$ (chosen to be an isotropic Gaussian distribution) and running the reverse process. We highly encourage the readers to refer to Appendix A for a more detailed background on diffusion models.

## 3    DiffuseVAE: VAEs meet Diffusion Models

### 3.1    DiffuseVAE Training Objective

Given a high-resolution image $x_0$, an auxiliary conditioning signal $y$ to be modelled using a VAE, a latent representation $z$ associated with $y$, and a sequence of $T$ representations $x_{1:T}$ learned by a diffusion model, the DiffuseVAE joint distribution can be factorized as:

$$p(x_{0:T}, y, z) = p(z)p_\theta(y|z)p_\phi(x_{0:T}|y, z) \tag{7}$$

where $\theta$ and $\phi$ are the parameters of the VAE decoder and the reverse process of the conditional diffusion model, respectively. Furthermore, since the joint posterior $p(x_{1:T}, z|y, x_0)$ is intractable to compute, we approximate it using a surrogate posterior $q(x_{1:T}, z|y, x_0)$ which can also be factorized into the following conditional distributions:

$$q(x_{1:T}, z|y, x_0) = q_\psi(z|y, x_0)q(x_{1:T}|y, z, x_0) \tag{8}$$

where $\psi$ are the parameters of the VAE recognition network ($q_\psi(z|y, x_0)$). As considered in previous works (Sohl-Dickstein et al., 2015; Ho et al., 2020) we keep the DDPM forward process ($q(x_{1:T}|y, z, x_0)$) non-trainable throughout training. The log-likelihood of the training data can then be obtained as:

$$\log p(x_0, y) = \log \int p(x_{0:T}, y, z)dx_{1:T}dz \tag{9}$$

Since this estimate is intractable to estimate analytically, we optimize the ELBO corresponding to the log-likelihood. It can be shown that the log-likelihood estimate of the data can be approximated using the following lower bound (See Appendix D.1 for the proof)

$$\log p(x_0, y) \geq \underbrace{\mathbb{E}_{q_\psi(z|y,x_0)}[p_\theta(y|z)] - \mathcal{D}_{KL}(q_\psi(z|y, x_0)||p(z))}_{\mathcal{L}_{\text{VAE}}} +$$

$$\mathbb{E}_{z \sim q(z|y,x_0)}\left[\underbrace{\mathbb{E}_{q(x_{1:T}|y,z,x_0)}\left[\frac{p_\phi(x_{0:T}|y, z)}{q(x_{1:T}|y, z, x_0)}\right]}_{\mathcal{L}_{\text{DDPM}}}\right] \tag{10}$$

We next discuss the choice of the conditioning signal $y$, some simplifying design choices and several parameterization choices for the VAE and the DDPM models.

### 3.2 Simplifying design choices

In this work we are interested in unconditional modeling of data. To this end, we make the following simplifying design choices:

1. **Choice of the conditioning signal** $y$: We assume the conditioning signal $y$ to be $x_0$ itself which ensures a deterministic mapping between $y$ and $x_0$. Given this choice, we do not condition the reverse diffusion process on $y$ and take it as $p_\phi(x_{0:T}|z)$ in Eq. 10.

2. **Choice of the conditioning signal** $z$: Secondly, instead of conditioning the reverse diffusion directly on the VAE inferred latent code $z$, we condition the second stage DDPM model on the VAE reconstruction $\hat{x}_0$ which is a deterministic function of $z$.

3. **Two-stage training**: We train Eq. 10 in a sequential two-stage manner, i.e., first optimizing $\mathcal{L}_{\text{VAE}}$ and then optimizing for $\mathcal{L}_{\text{DDPM}}$ in the second stage while fixing $\theta$ and $\psi$ (i.e. freezing the VAE encoder and the decoder).

With these design choices, as shown in Fig. 2, the DiffuseVAE training objective reduces to simply training a VAE model on the training data $x_0$ in the first stage and conditioning the DDPM model on the VAE reconstructions in the second stage. We next discuss the specific parameterization choices for the VAE and DDPM models. We also justify these design choices in Appendix E.

### 3.3 VAE parameterization

In this work, we only consider the standard VAE (with a single stochastic layer) as discussed in Section 2.1. However, in principle, due to the flexibility of the DiffuseVAE two-stage training, more sophisticated, multi-stage VAE approaches as proposed in (Razavi et al., 2019; Child, 2021; Vahdat & Kautz, 2021) can also be utilized to model the input data $x_0$. One caveat of using multi-stage VAE approaches is that we might no longer have access to the useful low-dimensional representation of the data.

### 3.4 DDPM parameterization

In this section, we discuss the two types of conditional DDPM formulations considered in this work.

#### 3.4.1 Formulation 1

In this formulation, we make the following simplifying assumptions

1. The forward process transitions are conditionally independent of the VAE reconstructions $\hat{x}$ and the latent code information $z$ i.e. $q(x_{1:T}|z, x_0) \approx q(x_{1:T}|x_0)$.

2. The reverse process transitions are conditionally dependent on only the VAE reconstruction, i.e., $p(x_{0:T}|z) \approx p(x_{0:T}|\hat{x}_0)$

A similar parameterization has been considered in recent work on conditional DDPM models (Ho et al., 2021; Saharia et al., 2021). We concatenate the VAE reconstruction to the reverse process representation $x_t$ at each time step $t$ to obtain $x_{t-1}$.

#### 3.4.2 Formulation 2

In this formulation, we make the following simplifying assumptions

1. The forward process transitions are conditionally dependent on the VAE reconstruction, i.e., $q(x_{1:T}|z, x_0) \approx q(x_{1:T}|\hat{x}_0, x_0)$

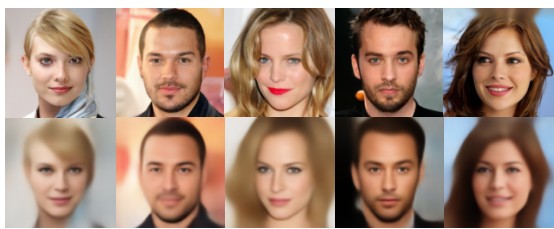
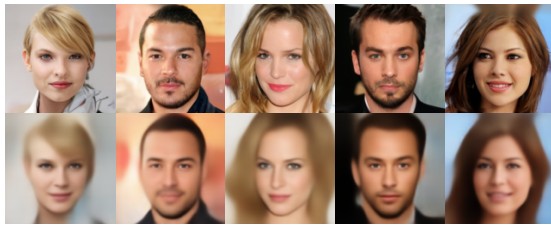

(a) Formulation-1     (b) Formulation-2

Figure 3: Illustration of the generator-refiner framework in DiffuseVAE. The VAE generated samples (Bottom row) are refined by the Stage-2 DDPM model with T=1000 during inference (Top Row).

2. The reverse process transitions are conditionally dependent on only the VAE reconstruction, i.e., $p(x_{0:T}|z) \approx p(x_{0:T}|\hat{x}_0)$

Specifically, we design the forward process transitions to incorporate the VAE reconstruction $\hat{x}_0$ as follows:

$$q(x_1|x_0, \hat{x}_0) = \mathcal{N}(\sqrt{1-\beta_1}x_0 + \hat{x}_0, \beta_1 I) \tag{11}$$

$$q(x_t|x_{t-1}, \hat{x}_0) = \mathcal{N}(\sqrt{1-\beta_t}x_{t-1} + (1-\sqrt{1-\beta_t})\hat{x}_0, \beta_t I) \qquad \text{for} \quad t > 1$$

It can be shown that the forward conditional marginal in this case becomes (See Appendix D.2 for proof)

$$q(x_t|x_0, \hat{x}_0) = \mathcal{N}(\sqrt{\bar{\alpha}_t}x_0 + \hat{x}_0, (1-\bar{\alpha}_t)I) \tag{12}$$

For $t = T$ and a *well-behaved* noise schedule $\beta_t$, $\bar{\alpha}_T \approx 0$ which implies $q(x_T|x_0, \hat{x}_0) \approx \mathcal{N}(\hat{x}_0, I)$. Intuitively, this means that the Gaussian $\mathcal{N}(\hat{x}_0, I)$ becomes our base measure ($p(x_T)$) during inference on which we need to run our reverse process. Since the simplified denoising training formulation proposed in (Ho et al., 2020) depends on the functional form of the forward process posterior $q(x_{t-1}|x_t, x_0)$, this formulation results in several modifications in the standard DDPM training and inference which are discussed in Appendix B.

## 4 Experiments

We now investigate several properties of the DiffuseVAE model. We use a mix of qualitative and quantitative evaluations for demonstrating these properties on several image synthesis benchmarks including CIFAR-10 (Krizhevsky, 2009), CelebA-64 (Liu et al., 2015), CelebA-HQ (Karras et al., 2018) and LHQ-256 (Skorokhodov et al., 2021) datasets. For quantitative evaluations involving sample quality, we use the FID (Heusel et al., 2018) metric. We also report the Inception Score (IS) metric (Salimans et al., 2016) for state-of-the-art comparisons on CIFAR-10. For all the experiments, we set the number of diffusion time-steps ($T$) to 1000 during training. The noise schedule in the DDPM forward process was set to a linear schedule between $\beta_1 = 10^{-4}$ and $\beta_2 = 0.02$ during training. More details regarding the model and training hyperparameters can be found in Appendix F. Some additional experimental results are presented in Appendix G.

### 4.1 Generator-refiner framework

Fig. 3 shows samples generated from the proposed DiffuseVAE model trained on the CelebA-HQ dataset at the 128 x 128 resolution and their corresponding Stage-1 VAE samples. For both DiffuseVAE formulations-1 and 2, DiffuseVAE generated samples (Fig. 3 (Top row) are a refinement of the *blurry* samples generated by our single-stage VAE model (Bottom row).

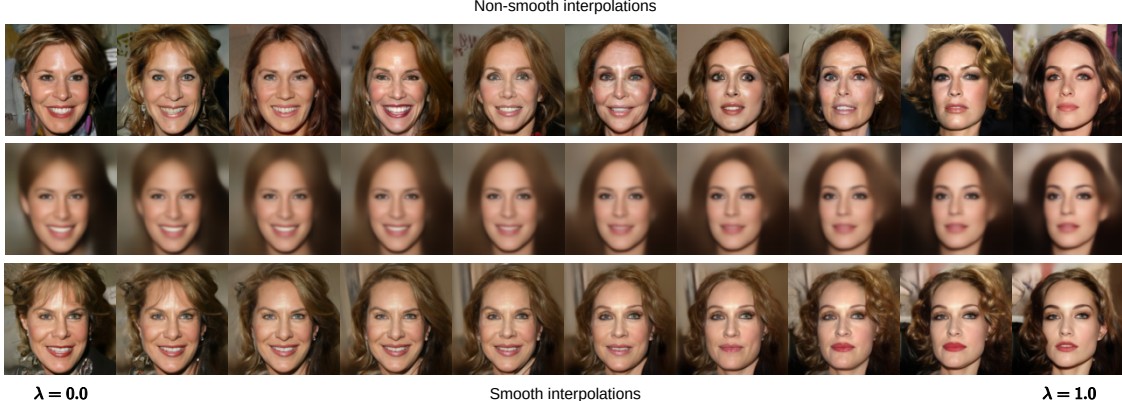

Figure 4: DiffuseVAE samples generated by linearly interpolating in the VAE latent space (Formulation-1, T=1000). $\lambda$ denotes the interpolation factor. *Middle row:* VAE generated interpolation between two samples. *Top row:* Corresponding DDPM refinements for VAE samples in the Middle Row. *Bottom row:* DDPM refinements for VAE samples in the Middle Row with shared DDPM stochasticity among all samples.

|  | FID@10k ↓ |
| --- | --- |
| Baseline VAE | 87.28 |
| Baseline VAE + DDPM Refiner (Form-1) | **10.87** |
| Baseline VAE + DDPM Refiner (Form-2) | **11.44** |

Table 1: Quantitative Illustration of the generator-refiner framework in DiffuseVAE for the CelebA-HQ (128 x 128) dataset. FID reported on 10k samples (Lower is better)

This observation qualitatively validates our *generator-refiner* framework in which the Stage-1 VAE model acts as a generator and the Stage-2 DDPM model acts as a refiner. The results in Table 1 quantitatively justify this argument where on the CelebA-HQ-128 benchmark, DiffuseVAE improves the FID score of a baseline VAE by about eight times. Additional qualitative results demonstrating this observation can be found in Fig. 13.

## 4.2 Controllable synthesis via low-dimensional DiffuseVAE latents

### 4.2.1 DiffuseVAE Interpolation

The proposed DiffuseVAE model consists of two types of latent representations: the low-dimensional VAE latent code $z_{vae}$ and the DDPM intermediate representations $x_{1:T}$ associated with the DDPM reverse process (which are of the same size of the input image $x_0$ and thus might not be beneficial for downstream tasks). We next discuss the effects of manipulating both $z_{vae}$ and $x_T$. Although, it is possible to inspect interpolations on the intermediate DDPM representations $x_{1:T-1}$, we do not investigate this case in this work. We consider the following interpolation settings:

**Interpolation in the VAE latent space** $z_{vae}$: We first sample two VAE latent codes $z_{vae}^{(1)}$ and $z_{vae}^{(2)}$ using the standard Gaussian distribution. We then perform linear interpolation between $z_{vae}^{(1)}$ and $z_{vae}^{(2)}$ to obtain intermediate VAE latent codes $\tilde{z}_{vae} = \lambda z_{vae}^{(1)} + (1 - \lambda)z_{vae}^{(2)}$ for $(0 < \lambda < 1)$, which are then used to generate the corresponding DiffuseVAE samples.

Fig. 4 (Middle Row) shows the VAE samples generated by interpolating between two sampled VAE codes as described previously. The corresponding DiffuseVAE generated samples obtained by interpolating in the $z_{vae}$ space are shown in Fig. 4 (Top Row). It can be observed that the refined samples corresponding to the blurry VAE samples preserve the overall structure of the image (facial expressions, hair style, gender etc).

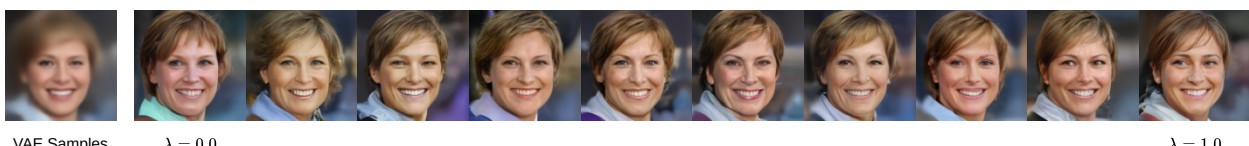

VAE Samples    $\lambda = 0.0$                                                                                                          $\lambda = 1.0$

Figure 5: DiffuseVAE samples generated by linearly interpolating in the $x_T$ latent space (Formulation-1, T=1000). $\lambda$ denotes the interpolation factor.

However, due to the stochasticity in the reverse process sampling in the second stage DDPM model, minor image details (like lip color and minor changes in skin tone) do not vary smoothly between the interpolation samples due to which the overall interpolation is not smooth. This becomes more clear when interpolating the DDPM latent $x_T$ while keeping the VAE code $z_{vae}$ fixed as discussed next.

**Interpolation in the DDPM latent space with fixed** $z_{vae}$: Next, we sample the VAE latent code $z_{vae}$ using the standard Gaussian distribution. With a fixed $z_{vae}$, we then sample two initial DDPM representations $x_T^{(1)}$ and $x_T^{(2)}$ from the reverse process base measure $p(x_T)$. We then perform linear interpolation between $x_T^{(1)}$ and $x_T^{(2)}$ with a fixed $z_{vae}$ to generate the final DiffuseVAE samples (Note that interpolation is not performed on other DDPM latents, $x_{1:T}$, which are obtained using ancestral sampling from the corresponding $x_T$'s as usual).

Fig 5 shows the DiffuseVAE generated samples with a fixed $z_{vae}$ and the interpolated $x_T$. As can be observed, interpolating in the DDPM latent space leads to changes in minor features (skin tone, lip color, collar color etc.) of the generated samples while major image structure (face orientation, gender, facial expressions) is preserved across samples. This observation implies that the low-dimensional VAE latent code mostly controls the structure and diversity of the generated samples and has more entropy than the DDPM representations $x_T$, which carry minor stochastic information. Moreover, this results in non-smooth DiffuseVAE interpolations. We discuss a potential remedy next.

**Handling the DDPM stochasticity**: The stochasticity in the second stage DDPM sampling process can occasionally result in artifacts in DiffuseVAE samples which might be undesirable in downstream applications. To make the samples generated from DiffuseVAE deterministic (i.e. controllable only from $z_{vae}$), we simply share all stochasticity in the DDPM reverse process (i.e. due to $x_T$ and $z_t$) across all generated samples. This simple technique adds more consistency in our latent interpolations as can be observed in Fig. 4 (Bottom Row) while also enabling deterministic sampling. This observation is intuitive as initializing the second stage DDPM in DiffuseVAE with different stochastic noise codes during sampling might be understood as imparting different styles to the refined sample. Thus, sharing this stochasticity in DDPM sampling across samples implies using the same stylization for all refined samples leading to smoothness between interpolations. Having achieved more consistency in our interpolations, we can now utilize the low-dimensional VAE latent code for controllable synthesis which we discuss next.

### 4.2.2   From Interpolation to Controllable Generation

Since DiffuseVAE gives us access to the entire low dimensional VAE latent space, we can perform image manipulation by performing vector arithmetic in the VAE latent space (See Appendix G.2 for details). The resulting latent code can then be used to sample from DiffuseVAE to obtain a refined manipulated image. As discussed in the previous section, we share the DDPM latents across samples to prevent the generated samples from using different styles. Fig. 6 demonstrates single-attribute image manipulation using DiffuseVAE on several attributes like *Gender*, *Age* and *Hair texture*. Moreover, the vector arithmetic in the latent space can be composed to generate composite edits (See Fig. 6), thus signifying the usefulness of a low-dimensional latent code representation. Some additional results on image manipulation are illustrated in Fig. 14.

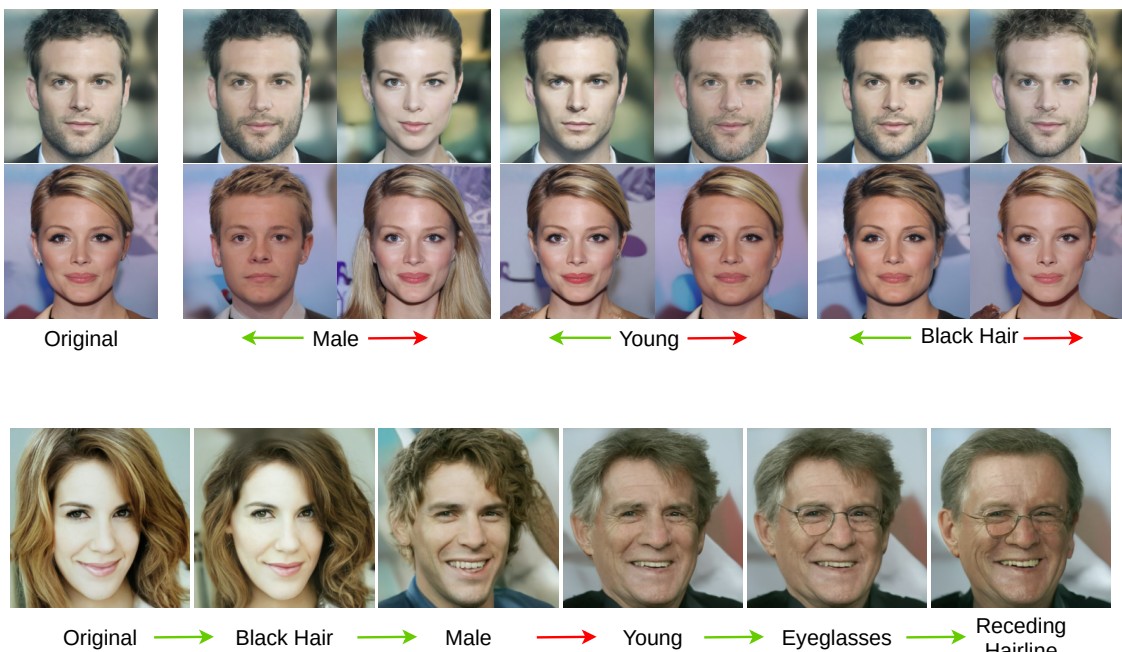

Figure 6: Controllable generation on DiffuseVAE generated samples on the CelebA-HQ 256 dataset. Red and green arrows indicate vector subtract and addition operations respectively. Top and Bottom panels show single edits and composite edits respectively.

### 4.3 Better Sampling Speed vs Quality tradeoffs with DiffuseVAE

There exists a trade-off between the number of reverse process sampling steps vs the quality of the generated samples in DDPMs. Usually the best sample quality is achieved when the number of reverse process steps used during inference matches the number of time-steps used during training. However, this can be very time-consuming (Song et al., 2021a). On the other hand, as the number of reverse process steps is reduced, the sample quality gets worse. We next examine this trade-off in detail.

**Comparison with a baseline unconditional DDPM**: Table 2 compares the sample quality (in terms of FID) vs the number of sampling steps between DiffuseVAE and our unconditional DDPM baseline on the CelebA-HQ-128 dataset. For all time-steps $T = 10$ to $T = 100$, DiffuseVAE outperforms the standard DDPM by large margins in terms of FID. Between DiffuseVAE formulations, the sample quality is similar with Formulation-1 performing slightly better. More notably, the FID score of DiffuseVAE at $T = 25$ and $50$ is better than that of unconditional DDPM at $T = 50$ and $100$ respectively. Thus, in low time-step regimes, the speed vs quality tradeoff in DiffuseVAE is significantly better than an unconditional DDPM baseline. It is worth noting that this property is intrinsic to DiffuseVAE as the model was not specifically trained to reduce the number of reverse process sampling steps during inference (Salimans & Ho, 2022).

However, at $T = 1000$ the unconditional DDPM baseline performs better than both DiffuseVAE formulations-1 and 2. We hypothesize that this gap in performance can be primarily attributed to the prior-hole problem, i.e., the mismatch between the VAE prior $p(z)$ and the aggregated posterior $q(z)$ (Bauer & Mnih, 2019; Dai & Wipf, 2019; Ghosh et al., 2020) due to which VAEs can generate poor samples from regions of the latent space unseen during training. DDPM refinement of such samples can affect the FID scores negatively. We confirm this hypothesis next.

**Improving DiffuseVAE sample quality using post-fitting**: One way to alleviate the prior-hole problem is to fit a density estimator (denoted by Ex-PDE) on the training latent codes and sample from this estimator during inference as in (van den Oord et al., 2017; Razavi et al., 2019; Ghosh et al., 2020). Along similar lines, we fit a GMM on the VAE latent code representations of the training data. We then use this estimator to

| | 10 | 25 | 50 | 100 | 1000 |
|---|---|---|---|---|---|
| DDPM (Uncond) | 41.25 | 27.83 | 21.40 | 16.29 | **8.93** |
| DiffuseVAE (Form-1) | 31.11 | **19.44** | **15.31** | **13.68** | 12.63 |
| DiffuseVAE (Form-2) | **31.08** | 19.67 | 15.96 | 13.96 | 13.20 |
| DiffuseVAE (Form-1, GMM=100) | 30.74 | **18.55** | **14.10** | **12.12** | 10.87 |
| DiffuseVAE (Form-2, GMM=100) | **30.66** | 18.98 | 14.45 | 12.50 | 11.44 |

Table 2: Comparison of sample quality (FID@10k) vs speed on the CelebA-HQ-128 dataset (DiffuseVAE vs unconditional DDPM). Top Row represents the number of reverse process sampling steps.

| | CelebAHQ-128 | | | | CelebA-64 | | | |
|---|---|---|---|---|---|---|---|---|
| | 10 | 25 | 50 | 100 | 10 | 25 | 50 | 100 |
| DDIM (uncond) | 34.36 | 25.04 | 19.83 | 16.69 | 14.14 | 7.88 | 6.77 | 6.38 |
| DiffuseVAE (Form-1) | 19.42 | 15.12 | 14.53 | 14.53 | 10.79 | 6.87 | 6.08 | 5.82 |
| DiffuseVAE (Form-1, Ex-PDE) | **18.01** | **13.21** | **12.40** | **12.28** | **10.44** | **6.59** | **5.81** | **5.55** |
| DiffuseVAE (Form-2) | 17.51 | 13.45 | 12.56 | 12.51 | 9.81 | 6.34 | 5.83 | 5.59 |
| DiffuseVAE (Form-2, Ex-PDE) | **16.47** | **11.62** | **10.83** | **10.28** | **9.56** | **5.90** | **5.43** | **5.21** |

Table 3: Comparison of sample quality (FID@10k) vs speed between DiffuseVAE and the unconditional DDIM on the CelebA-HQ-128 and CelebA-64 datasets. DiffuseVAE with Form-2 shows a better speed-quality tradeoff than Form-1. Overall, DiffuseVAE achieves upto 4x and 10x speedups on the CelebA-64 and the CelebA-HQ-128 datasets respectively as compared to the uncondtional DDIM

.

sample VAE latent codes during DiffuseVAE sampling. Table 2 shows the FID scores on the CelebA-HQ-128 dataset for both DiffuseVAE formulations using a GMM with 100 components. Across all time-steps, using Ex-PDE during sampling leads to a reduced gap in sample quality at $T = 1000$, thereby confirming our hypothesis. We believe that the remaining gap can be closed by using stronger density estimators which we do not explore in this work. Moreover, a side benefit of using a Ex-PDE during sampling is further improvement in the speed-quality tradeoff.

**Further improvements with DDIM**: DDIM (Song et al., 2021a) employs a non-Markovian forward process and achieves a better speed-quality tradeoff than DDPM along with deterministic sampling. Since DiffuseVAE employs a DDPM model in the refiner stage, we found DDIM sampling to be complementary with the DiffuseVAE framework. Notably, since the forward process for DiffuseVAE (Form-2) is different, we derive the DDIM updates for this formulation in Appendix B. Table 3 compares the speed-quality tradeoff between DDIM and DiffuseVAE (with DDIM sampling) on the CelebA-HQ-128 and CelebA-64 datasets. DiffuseVAE (both formulations) largely outperforms the standard unconditional DDIM at all time-steps. For the CelebA-HQ-128 benchmark, similar to our previous observation, DiffuseVAE (with DDIM sampling and Ex-PDE using GMMs) at $T = 25$ and 50 steps performs better than the standard DDIM at $T = 50$ and 100 steps respectively. In fact, at $T = 10$, DiffuseVAE (with Formulation-2) achieves a FID of 16.47 which is better than DDIM with $T = 100$ steps, thus providing a speedup of almost 10x. Similarly for the CelebA-64 benchmark, at $T = 25$, DiffuseVAE (Formulation-2) performs similarly to the unconditional DDIM at $T = 100$, thus providing a 4x speedup. Lastly, it can be observed from Tables 3, 11 and 12 that in the low time-step regime, DiffuseVAE (Form-2) usually performs better than Form-1 and that the speed-quality trade-off in DiffuseVAE becomes better with increasing image resolutions.

## 4.4 State-of-the-art comparisons

For reporting comparisons with the state-of-the-art we primarily use the FID (Heusel et al., 2018) metric to assess sample quality. We compute FID on 50k samples for CIFAR-10 and CelebA-64. For comparisons on the CelebA-HQ-256 dataset, we report the FID only for 10k samples (as opposed to 30k samples which is the norm on this benchmark) due to compute limitations. Due to this, we anticipate the true FID score on this benchmark using our method to be lower. However, as we show, the FID score obtained by DiffuseVAE on this benchmark on 10k samples is still comparable to state-of-the-art.

|  | Method | FID@50k ↓ | IS ↑ |
|---|---|---|---|
| **Ours** | DiffuseVAE (Form-1, T=1000) | 2.95 | 9.60 ± 0.11 |
|  | DiffuseVAE (Form-2, T=1000) | 2.86 | 9.59 ± 0.13 |
|  | DiffuseVAE (Form-1, T=1000, GMM=50) | 2.84 | 9.51 ± 0.08 |
|  | DiffuseVAE (Form-2, T=1000, GMM=50) | 2.80 | 9.51 ± 0.08 |
|  | DiffuseVAE-72M (Form-2, T=1000, GMM=50) | 2.62 | 9.75 ± 0.08 |
|  | DDPM (T=1000, Our impl.) | 3.01 | 9.55 ± 0.16 |
|  | VAE Baseline | 139.50 | 3.23 ± 0.02 |
|  | VAE Baseline (GMM=50) | 137.68 | 3.30 ± 0.02 |
| **VAE-based methods** | VAEBM (Xiao et al., 2021) (w/ PC) | 12.19 | 8.43 |
|  | DC-VAE (Parmar et al., 2021) | 17.90 | 8.2 |
|  | NVAE (Vahdat & Kautz, 2021) | 51.67 | 5.51 |
|  | NCP-VAE (Aneja et al., 2020) | 24.08 | - |
|  | LSGM (FID) (Vahdat et al., 2021) | 2.10 | - |
|  | D2C (Sinha et al., 2021) | 10.15 | - |
| **GAN-based methods** | AutoGAN (Cao et al., 2020) | 12.4 | 8.55 ± 0.1 |
|  | ProGAN (Karras et al., 2018) | 15.52 | 8.56 ± 0.10 |
|  | StyleGAN2 (w/o ADA) (Karras et al., 2019) | 8.32 | 9.21 ± 0.09 |
|  | StyleGAN2-ADA (Karras et al., 2020a) | 2.92 | 9.83 ± 0.04 |
|  | SNGAN (Miyato et al., 2018) | 21.7 | 8.22 ± 0.05 |
|  | SNGAN + DDLS (Che et al., 2021) | 15.42 | 9.09 ± 0.10 |
| **Score-based methods** | NCSN (Song & Ermon, 2020a) | 25.32 | 8.87 ± 0.12 |
|  | NCSNv2 (w/denoising) (Song & Ermon, 2020b) | 10.87 | 8.40 ± 0.07 |
|  | DDPM (Ho et al., 2020) | 3.17 | 9.46 ± 0.11 |
|  | SDE (NCSN++) (Song et al., 2021b) | 2.45 | 9.73 |
|  | SDE (DDPM++) (Song et al., 2021b) | 2.78 | 9.64 |

Table 4: Generative performance on unconditional CIFAR-10. FID and IS computed on 50k samples

Table 4 shows quantitative comparison between DiffuseVAE and other state-of-the-art unconditional generative models in terms of sample quality (FID@50k) and sample diversity (IS) on the CIFAR-10 dataset. Interestingly, our unconditional DDPM baseline achieves better FID scores on CIFAR-10 than reported in (Ho et al., 2020). DiffuseVAE clearly outperforms the DDPM baseline (with and without Ex-PDE) in terms of FID while maintaining a competitive IS score with continuous score based methods indicating good sample diversity. Notably, with the exception of LSGM (Vahdat et al., 2021), DiffuseVAE outperforms all prior state-of-the-art VAE-based methods (Vahdat & Kautz, 2021; Xiao et al., 2021; Sinha et al., 2021), even when most of these methods utilize powerful hierarchical VAE-based backbones. In contrast, DiffuseVAE utilizes a simple VAE backbone with a very poor baseline FID score and it would be interesting to benchmark LSGM using a simple VAE backbone as ours (some initial evaluations on CIFAR-10 already suggest that LSGM might perform much worse than DiffuseVAE with a simple VAE baseline [1]). In this work our CIFAR-10 model is the same size as in (Ho et al., 2020) which is an order of magnitude smaller than LSGM (See Table 14). Indeed, like LSGM, DiffuseVAE can also take advantage of larger model sizes (DiffuseVAE-72M with Ex-PDE achieves a FID of **2.62** and a mean IS of **9.75** on CIFAR-10. See Appendix G.4). *Moreover, to the best of our knowledge, DiffuseVAE is the first model to outperform StyleGAN2-ADA (Karras et al., 2020a) on this benchmark while being trained using non-adversarial losses and retaining access to a low-dimensional latent code.*

We also benchmarked DiffuseVAE (with Ex-PDE) on two popular face image benchmarks: CelebA-64 and CelebA-HQ-256. On the CelebA-64 benchmark, DiffuseVAE performs comparably with the DDPM baseline. Similar to CIFAR10, DiffuseVAE outperforms other VAE-based methods (Sinha et al., 2021; Aneja et al., 2020; Xiao et al., 2021) by a significant margin. We observed similar trends on the CelebA-HQ-256 dataset where DiffuseVAE outperforms competing VAE based methods except LSGM and is comparable to VQGAN (Esser et al., 2020). However, when comparing with LSGM on this benchmark, similar arguments as pointed out for CIFAR-10 hold. Interestingly, we found that for CelebA-HQ-256 dataset, samples generated during intermediate training stages (and even after convergence) suffer from color bleeding. We found that this problem can be alleviated by using temperature sampling in the second stage DDPM latents (Appendix G.4). Therefore, only for $T = 1000$, we report the FID scores on this benchmark with a scaling factor of 0.8.

---

[1]See https://openreview.net/forum?id=P9TYGOj-wtG&noteId=Z7AYukcBJ_q

| Method | FID@50k ↓ |
|---|---|
| DiffuseVAE (Form-1, T=1000, GMM=75) | 4.05 |
| DiffuseVAE (Form-2, T=1000, GMM=75) | 3.97 |
| DDPM (T=1000, Our impl.) | 3.93 |
| VAE Baseline (GMM=75) | 72.11 |
| D2C (Sinha et al., 2021) | 5.7 |
| NCP-VAE (Aneja et al., 2020) | 5.25 |
| VAEBM (Xiao et al., 2021) | 5.31 |
| NVAE (Vahdat & Kautz, 2021) | 14.74 |
| NCSN (Song & Ermon, 2020a) | 25.30 |
| NCSNv2 (Song & Ermon, 2020b) | 10.23 |
| QA-GAN (PARIMALA & Channappayya, 2019) | 6.42 |
| COCO-GAN (Lin et al., 2020) | 4.0 |

Table 5: Generative performance on CelebA-64

| Method | FID ↓ |
|---|---|
| DiffuseVAE (T=1000, GMM=100, FID@10k) | 11.28 |
| VAE Baseline (GMM=100, FID@10k) | 97.07 |
| LSGM (Vahdat et al., 2021) | 7.22 |
| VQGAN + Transformer (Esser et al., 2020) | 10.2 |
| D2C (Sinha et al., 2021) | 18.74 |
| DCVAE (Parmar et al., 2021) | 15.81 |
| VAEBM (Xiao et al., 2021) | 20.38 |
| NCP-VAE (Aneja et al., 2020) | 24.8 |
| NVAE (Vahdat & Kautz, 2021) | 40.26 |

Table 6: Generative performance on CelebA-HQ-256

## 4.5 Generalization to different noise types

To test if DiffuseVAE can generalize over different types of noisy conditioning signals during sample generation, we condition the second stage DDPM model in DiffuseVAE (pre-trained on the CIFAR-10 dataset) on different types of noisy conditioning signals (instead of the VAE reconstruction). More specifically, we experiment with two such types of conditioning signals obtained by adding noise to CIFAR-10 test samples: downsampling CIFAR samples to 16x16 resolution (effectively blurring them when scaled back) and adding Gaussian noise (with standard deviation = 0.3). Final DiffuseVAE samples obtained after conditioning on these noisy inputs are visualized in Fig. 7 (with additional results on the CelebA-HQ-128 samples illustrated in Fig. 16). We observed that DiffuseVAE is able to recover the original samples from the noisy inputs which demonstrates generalization to different noisy conditioning inputs.

Intuitively, these results can be expected since, during training, the proposed DiffuseVAE method learns to refine VAE reconstructions which lack a lot of detail. Hence the task of refining these reconstructions might be more challenging, thus allowing the network to generalize to *simpler* tasks inherently as illustrated above. However, it is worth noting that certain artifacts in the generated refinements are evident (For instance in Figure 16, the sample quality shows a sharp degradation as more noise is added to the conditioning signal), leaving scope for design of more stronger conditioning mechanisms in diffusion models that allow to adapt conditional diffusion models on downstream tasks like image super-resolution in an out-of-the-box fashion.

## 5 Related Work

Following the seminal work of (Sohl-Dickstein et al., 2015; Ho et al., 2020) on diffusion models, there has been a lot of recent progress in both unconditional (Nichol & Dhariwal, 2021; Dhariwal & Nichol, 2021; Kingma et al., 2021) and conditional diffusion models (Ho et al., 2021; Saharia et al., 2021; Choi et al., 2021; Chen et al., 2020) (including score-based models (Song et al., 2021b; Song & Ermon, 2020a)) for a variety of downstream tasks including image synthesis, audio synthesis and likelihood estimation among others. Here we only compare DiffuseVAE to recent methods which attempt to combine VAEs with diffusion models. We refer the readers to Appendix C for a detailed comparison of DiffuseVAE to other types of model families.

Among recent advances, there are several works which apply diffusion models in the latent space of powerful autoencoding baselines. D2C (Sinha et al., 2021) utilizes a learned diffusion-based prior over the NVAE (Vahdat & Kautz, 2021) latent representations while also refining the latent space using a contrastive loss. LSGM (Vahdat et al., 2021) performs score-based generative modeling in the latent space of NVAE baseline. Similarly, Latent Diffusion Models (LDM) (Rombach et al., 2021) apply diffusion models in the latent space of a powerful pretrained VQ-GAN (Esser et al., 2020) autoencoding baseline. In contrast, our method refines "blurry" reconstructions generated by an extremely lightweight VAE using a downstream diffusion model. A possible benefit of having a generator-refiner framework in contrast to the latent diffusion framework could be the requirement of a powerful VAE baseline as a pre-requisite to generate high-quality samples. Since there exists a trade-off between latent code disentanglement and high quality reconstructions (Higgins et al., 2017), the need of a high fidelity autoencoding baseline can be disadvantageous in situations where a fine-grained

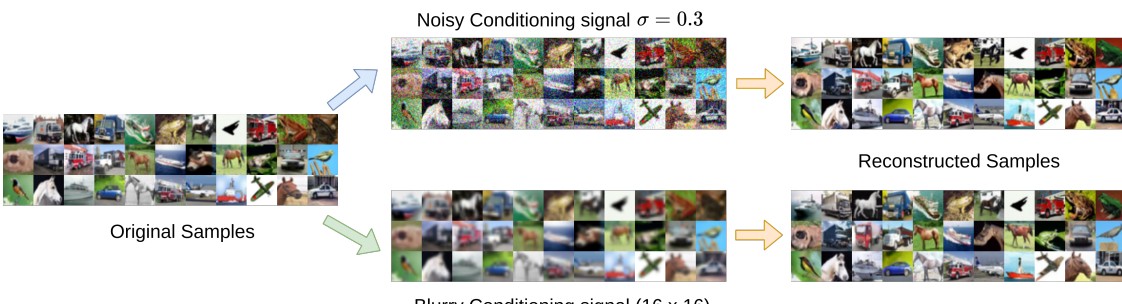

Figure 7: Illustration of DiffuseVAE generalization to different noise types in the conditioning signal on the CIFAR-10 test set.

control over the generated samples is required. We hypothesize that this problem is alleviated in DiffuseVAE since our first stage model can readily tradeoff more disentanglement for lower fidelity reconstructions due to a powerful second stage diffusion-based refiner model. Lastly, we hypothesize that the latent diffusion framework is complementary to DiffuseVAE since the prior used in our VAE training can be modeled using a diffusion model.

(Luo & Hu, 2021) present a probabilistic autoencoding framework for point cloud generation via a VAE-like encoder and a diffusion model based decoder. Notably, the most closest to our approach is the concurrent work on DiffAE (Preechakul et al., 2022) which uses an end-to-end autoencoding framework for conditioning the diffusion process decoder on the latent code output of an encoder. This equips the diffusion model with a low-dimensional latent space. However, since the model is non-probabilistic, DiffAE relies on fitting a powerful DDIM density estimator on the latent space of the encoder to enable sampling. Moreover, it's unclear if DiffAE exhibits good sample quality when fitting simple density estimators on the encoder latent space. In contrast, sampling in DiffuseVAE is straightforward due to a probabilistic formulation. Additionally, DiffuseVAE can also take advantage of fitting external density estimators on the latent space as demonstrated in this work.

## 6 Limitations and Discussion

In this work, we presented a novel unifying framework for training VAEs and diffusion models and demonstrated its effectiveness in generating high-quality samples, providing a better sample quality vs number of steps trade-off while equipping DDPM with a low dimensional latent code which can be used for controllable synthesis using DDPM, and generalizing to different types of noise in the conditioning signal. However, the DiffuseVAE model is not without its limitations:

1. Due to a generator-refiner framework, the semantics of the final generated samples depends largely on the coarse sample generated by the *generator* model (a simple VAE in our case). Therefore, if the coarse sample is not semantically meaningful, this will propagate to the final generated sample after refinement. This can be expected from VAEs due to a mismatch between the aggregated posterior $q(z)$ and the prior $p(z)$ during VAE training which we alleviate using Ex-PDE estimation but the problem still persists (which is evident from the gap in sample quality between an unconditional DDPM baseline and DiffuseVAE even after Ex-PDE).

2. We also observed that when the conditioning signal provided by the first stage VAE is uninformative (too blurry), the second stage DDPM model can generate unpredictable refinements. On this note, it would be interesting to explore the impact of the choice of VAE on the overall sample quality of the model. Moreover, since we work with vanilla VAEs, some artifacts in controllable synthesis results are evident due to correlated attribute-specific latent directions (See Figure 15). Using variants like $\beta$-VAEs (Higgins et al., 2017) can help achieve more disentanglement between image attributes leading to better controllable synthesis results.

3. In this work, since we focus on sample quality, we did not explore the impact of the diffusion model training on the latent space of the VAE when trained end-to-end. It would be interesting to explore if end-to-end training might alleviate some problems with VAE's.

4. Lastly, it would be interesting to explore stronger conditioning mechanisms in the context of diffusion models which reduce the reliance of the final sample on the stochastic DDPM sub-code. In the context of DiffuseVAE, this can also be useful in improving model generalization to downstream tasks like image super-resolution and denoising as presented in Section 4.5

## Broader Impact Statement

In addition to modelling images, our proposed approach can also be used to model data of other modalities like speech, text, etc. It has the potential to mitigate bias and privacy issues for related ML models that require data collection and annotation. However, such techniques could also be misused to produce fake or misleading information, and researchers should be aware of these risks and explore the proposed approaches responsibly.

### Acknowledgments

We would like to thank Ben Poole for his insightful comments and suggestions through the course of this project. We would also like to thank Google Cloud for supporting our research in the form of cloud compute credits.

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

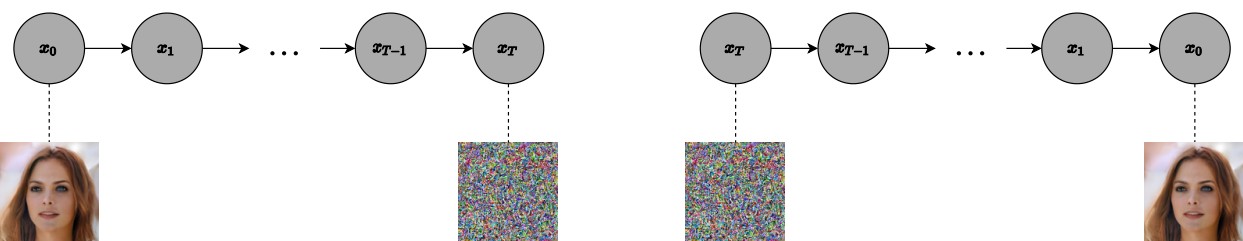

Figure 8: Forward Process                 Figure 9: Reverse Process

## A   Background on Diffusion models

DDPMs (Sohl-Dickstein et al., 2015; Ho et al., 2020) are latent-variable models consisting of a forward noising process $(q(x_{1:T}|x_0))$ (corresponding to an inference model in other generative model families like VAEs (Kingma & Welling, 2014; Rezende & Mohamed, 2016). See Fig. 8) and a reverse denoising process $(p(x_{0:T}))$ (corresponding to a generator or decoder in VAEs. See Fig. 9). The forward process is modeled using a Markov chain which gradually destroys the structure of the data $x_0$ over a number of time-steps T. Similarly, the reverse process is also modeled as a Markov chain which learns to recover the original data $x_0$ from the noisy input $x_T$. The form of the forward process and some notable properties of the forward process conditional distributions are summarized in the following equations ( Eqs. (13-19)).

$$q(x_{1:T}|x_0) = \prod_{t=1}^{T} q(x_t|x_{t-1}) \tag{13}$$

$$q(x_t|x_{t-1}) = \mathcal{N}(\sqrt{1-\beta_t}x_{t-1}, \beta_t I) \tag{14}$$

The forward process of DDPMs admits a closed form for $x_t$ for any $t$, as follows:

$$q(x_t|x_0) = \mathcal{N}(\sqrt{\bar{\alpha}_t}x_0, (1-\bar{\alpha}_t)I) \tag{15}$$

$$\text{where } \alpha_t = (1-\beta_t) \text{ and } \bar{\alpha}_t = \prod_t \alpha_t \tag{16}$$

The forward process posteriors are also tractable and are given by

$$q(x_{t-1}|x_t, x_0) = \mathcal{N}(\tilde{\mu}_t(x_t, x_0), \tilde{\beta}_t) \tag{17}$$

$$\text{where } \tilde{\mu}_t(x_t, x_0) = \frac{\sqrt{\bar{\alpha}_{t-1}}\beta_t}{1-\bar{\alpha}_t}x_0 + \frac{\sqrt{\alpha_t}(1-\bar{\alpha}_{t-1})}{1-\bar{\alpha}_t}x_t \tag{18}$$

$$\text{and } \tilde{\beta}_t = \frac{1-\bar{\alpha}_{t-1}}{1-\bar{\alpha}_t}\beta_t \tag{19}$$

The reverse process can also be parameterized using a first-order Markov chain with a learned Gaussian transition distribution as follows

$$p(x_{0:T}) = p(x_T) \prod_{t=1}^{T} p_\theta(x_{t-1}|x_t) \tag{20}$$

$$p_\theta(x_{t-1}|x_t) = \mathcal{N}(\mu_\theta(x_t, t), \Sigma_\theta(x_t, t)) \tag{21}$$

$$p_\theta(x_{t-1}|x_t) = \mathcal{N}(\mu_\theta(x_t, t), \Sigma_\theta(x_t, t)) \tag{22}$$

Given a large enough $T$ and a well-behaved variance schedule of $\beta_t$, the distribution $q(x_T|x_0)$ will approximate an isotropic Gaussian. We can generate a new sample from the underlying data distribution $q(x_0)$ by sampling a latent from $p(x_T)$ (chosen to be an isotropic Gaussian distribution) and running the reverse process. As

proposed in (Ho et al., 2020), the reverse process in DDPM is trained to minimize the following upper bound over the negative log-likelihood (See (Sohl-Dickstein et al., 2015) for detailed proofs):

$$\mathbb{E}_q\left[\mathcal{D}_{KL}(q(x_T|x_0)\|p(x_T)) + \sum_{t>1}\mathcal{D}_{KL}(q(x_{t-1}|x_t,x_0)\|p_\theta(x_{t-1}|x_t)) - \log p_\theta(x_0|x_1)\right] \tag{23}$$

A notable aspect of the above objective is that all the KL divergences involve Gaussians and, consequently, are available in closed form. Notably, (Ho et al., 2020) parameterize the reverse process conditional $p_\theta(x_{t-1}|x_t)$ using the forward process posterior $q(x_{t-1}|x_t,x_0)$. (Ho et al., 2020) show that such a parameterization simplifies the second term in Eq. 23 at any given time-step $t$ to the following objective in Eq. 24.

$$\|\epsilon - \epsilon_\theta(\sqrt{\bar{\alpha}_t}x_0 + \sqrt{1-\bar{\alpha}_t}\epsilon, t))\|_2^2 \tag{24}$$

where $x_t = \sqrt{\bar{\alpha}_t}x_0 + \epsilon\sqrt{1-\bar{\alpha}_t}$ and $\epsilon \sim \mathcal{N}(0, I)$. Intuitively, this means that the reverse process in DDPM is trained to predict the noise added to the input $x_0$ at any time-step $t$. We use this *simplified* training formulation throughout our work to train all proposed parameterizations of diffusion models as (Ho et al., 2020) show that this formulation yields superior sample quality than other forms of reverse process parameterizations. For further details on the exact training and inference processes, we encourage the readers to refer to (Ho et al., 2020).

# B   Discussion of DiffuseVAE (Formulation-2)

---

**Algorithm 1** DDPM Training (Form. 2)

**repeat**
    $x_0 \sim q(x_0)$
    $\hat{x}_0 = VAE(x_0)$
    $t \sim \text{Uniform}(\{1 \dots \text{T}\})$
    $\epsilon \sim \mathcal{N}(0, I)$
    Take gradient descent step on:
      $\nabla_\theta \|\epsilon - \epsilon_\theta(\sqrt{\bar{\alpha}_t}x_0 + \hat{x}_0 + \sqrt{1 - \bar{\alpha}_t}\epsilon, t, \hat{x}_0)\|^2$
**until** convergence

---

**Algorithm 2** DDPM Inference (Form. 2)

$z_{\text{vae}} \sim \mathcal{N}(0, I)$
$y = \text{VAEDEC}(z_{\text{vae}})$
$x_T \sim \mathcal{N}(y, I)$
**for** t = T **to** 1 **do**
    $z = \mathcal{N}(0, I)$, if $t > 1$ else 0
    $\hat{x}_0 = \frac{1}{\sqrt{\bar{\alpha}_t}}(x_t - y - \epsilon_\theta(x_t, y, t)\sqrt{1 - \bar{\alpha}_t})$
    $\hat{x}_{t-1} = \gamma_0 \hat{x}_0 + \gamma_1 x_t + \gamma_2 y$
    $x_{t-1} = \hat{x}_{t-1} + z\hat{\sigma}_t$
**end for**

---

return $x_0 - y$

---

The DDPM training objective proposed in (Ho et al., 2020), has the following form:

$$\mathbb{E}_q \left[ \underbrace{\mathcal{D}_{KL}\left(q(x_T|x_0)\|p(x_T)\right)}_{L_T} + \sum_{t>1} \underbrace{\mathcal{D}_{KL}(q(x_{t-1}|x_t, x_0)\|p_\theta(x_{t-1}|x_t))}_{L_{t-1}} - \underbrace{\log p_\theta(x_0|x_1)}_{L_0} \right] \tag{25}$$

## B.1   Reverse Process parameterization

Following (Ho et al., 2020), we parameterize the reverse process transition $p_\theta(x_{t-1}|x_t)$ using the functional form of the forward process posterior $q(x_{t-1}|x_t, x_0)$. For the DiffuseVAE formulation proposed in Section 3.4.2 in our paper, the forward process conditional distributions can be specified as:

$$q(x_t|x_{t-1}, \hat{x}_0) = \mathcal{N}\left(\sqrt{1 - \beta_t}x_{t-1} + (1 - \sqrt{1 - \beta_t})\hat{x}_0, \beta_t I\right) \quad \text{where} \ \ t > 1 \tag{26}$$

$$q(x_t|x_0, \hat{x}_0) = \mathcal{N}\left(\sqrt{\bar{\alpha}_t}x_0 + \hat{x}_0, (1 - \bar{\alpha}_t)I\right) \tag{27}$$

The posterior distribution $q(x_{t-1}|x_t, x_0, \hat{x}_0)$ will also be a Gaussian distribution with the following form:

$$q(x_{t-1}|x_t, x_0, \hat{x}_0) = \mathcal{N}(\hat{\mu}_t(x_t, x_0, \hat{x}_0), \hat{\beta}_t I) \tag{28}$$

where,

$$\hat{\mu}_t(x_t, x_0, \hat{x}_0) = \underbrace{\frac{\beta_t \sqrt{\bar{\alpha}_{t-1}}}{1 - \bar{\alpha}_t}x_0 + \frac{(1 - \bar{\alpha}_{t-1})\sqrt{\alpha_t}}{1 - \bar{\alpha}_t}x_t}_{\tilde{\mu}_t(x_t, x_0)} + \underbrace{(1 - \frac{(1 - \bar{\alpha}_{t-1})\sqrt{\alpha_t}}{1 - \bar{\alpha}_t})\hat{x}_0}_{\kappa} \tag{29}$$

$$\hat{\beta}_t = \frac{(1 - \bar{\alpha}_{t-1})}{1 - \bar{\alpha}_t}\beta_t \ \ \text{and} \ \ x_0 = \frac{1}{\sqrt{\bar{\alpha}_t}}(x_t - \hat{x}_0 - \epsilon\sqrt{1 - \bar{\alpha}_t}) \tag{30}$$

$$\text{where} \ \ \epsilon \sim \mathcal{N}(0, I)$$

Hence the forward process posterior in this DiffuseVAE formulation is a shifted version of the forward process posterior proposed in (Ho et al., 2020). Since the VAE reconstruction $\hat{x}_0$ for an image $x_0$ is constant during DDPM training, we can parameterize the reverse process posterior as $\hat{\mu}_\theta(x_t, x_0, \hat{x}_0, t) = \tilde{\mu}_\theta(x_t, x_0, t) + \kappa\hat{x}_0$. Additionally, we keep the variance of the reverse process conditional fixed and equal to $\hat{\beta}_t$ as proposed in (Ho et al., 2020). Since $L_{t-1} \propto \|\hat{\mu}_t(x_t, x_0, y) - \hat{\mu}_\theta(x_t, x_0, y, t)\|^2$, the DDPM training objective in our formulation remains unchanged from the simplified denoising score matching objective proposed in (Ho et al., 2020).

## B.2 Choice of the decoder, $L_0$

One possible choice for the decoder is to set $p_\theta(x_0|x_1)$ to be a discrete independent decoder derived from the Gaussian $\mathcal{N}(\hat{\mu}_\theta(x_1, \hat{x}_0, 1), \hat{\beta}_1 I)$ (Ho et al., 2020). However, at $t = 1$, we have $\hat{\mu}_\theta(x_1, \hat{x}_0, 1) = x_0(x_1, \hat{x}_0, \epsilon_\theta) + \hat{x}_0$. Therefore, to account for the VAE reconstruction bias in the final DDPM output, we set our decoder $p_\theta(x_0|x_1) = \mathcal{N}(\hat{\mu}_\theta(x_1, \hat{x}_0, 1) - \hat{x}_0, \hat{\beta}_1 I)$. Without using this adjustment, we found the final DDPM samples to be a bit blurry in our initial experiments. The final training and inference algorithms are summarized in Algorithms 1 and 2 respectively. In Algorithm 2, the coefficients $\gamma_0, \gamma_1$ and $\gamma_2$ denote the coefficients of the forward process posterior in Eqn. 29.

## B.3 Integration with DDIM

We now derive the updates for the DiffuseVAE formulation-2 when combined with DDIM sampling. Given the form of the forward process marginal as in Eqn. 27, we assume the following form of the forward process posterior:

$$q(x_{t-1}|x_t, \hat{x}_0, x_0) = \mathcal{N}(\mu_t, \sigma_t^2) \tag{31}$$

$$\mu_t = \sqrt{\bar{\alpha}_{t-1}} x_0 + \sqrt{1 - \bar{\alpha}_{t-1} - \sigma_t^2} \left[ \frac{x_t - \sqrt{\bar{\alpha}_t} x_0}{\sqrt{1 - \bar{\alpha}_t}} \right] + \kappa \hat{x}_0 \tag{32}$$

$$\sigma_t^2 = \eta \left[ \frac{1 - \bar{\alpha}_{t-1}}{1 - \bar{\alpha}_t} \right] \left[ 1 - \frac{\bar{\alpha}_t}{\bar{\alpha}_{t-1}} \right] \tag{33}$$

We now have,

$$q(x_{t-1}|x_0, \hat{x}_0) = \int q(x_{t-1}|x_t, x_0, \hat{x}_0) q(x_t|x_0, \hat{x}_0) dx_t \tag{34}$$

Since both the distributions within the integral are gaussians, the resulting marginal will also be a gaussian with the following form:

$$q(x_{t-1}|x_0, \hat{x}_0) = \mathcal{N}(\bar{\mu}_t, \bar{\sigma}_t^2) \tag{35}$$

$$\bar{\mu}_t = \sqrt{\bar{\alpha}_{t-1}} x_0 + \left[ \kappa + \frac{\sqrt{1 - \bar{\alpha}_{t-1} - \sigma_t^2}}{\sqrt{1 - \bar{\alpha}_t}} \right] \hat{x}_0 \tag{36}$$

$$\bar{\sigma}_t^2 = 1 - \bar{\alpha}_{t-1} \tag{37}$$

However, we already know the form of the marginal $q(x_{t-1}|x_0, \hat{x}_0)$ from Eqn. 27 as follows:

$$q(x_{t-1}|x_0, \hat{x}_0) = \mathcal{N}(\sqrt{\bar{\alpha}_{t-1}} x_0 + \hat{x}_0, 1 - \bar{\alpha}_{t-1} I) \tag{38}$$

Therefore it implies that,

$$\kappa = 1 - \frac{\sqrt{1 - \bar{\alpha}_{t-1} - \sigma_t^2}}{\sqrt{1 - \bar{\alpha}_t}} \tag{39}$$

This completes the analysis of the modified DDIM forward process posterior which is compatible with DiffuseVAE formualation-2

## B.4 Primary Intuition

The primary intuition behind constructing such a formulation is that by initializing the base distribution from a VAE reconstruction, we can hope to speed up the reverse diffusion process. In the low time-step regime, DiffuseVAE (Form-2) usually performs better than (Form-1) (See Tables 3, 11 and 12). These results indicate our hypothesis might hold valid in the low-time-step regime in diffusion models.

## C   Related Work

Recent work in DDPMs also includes improving the speed vs sample quality tradeoff in the DDPM sampling process (Song et al., 2021a; Watson et al., 2021; Luhman & Luhman, 2021; Salimans & Ho, 2022; Xiao et al., 2022). We consider these advances in speeding up diffusion models are complementary to our work and can also be used to improve the sampling efficiency of DiffuseVAE. However, on the contrary, a majority of such methods were designed for improving sampling speeds in DDPMs while DiffuseVAE improves this tradeoff inherently. Similarly for VAEs (Kingma & Welling, 2014; Rezende & Mohamed, 2016), there has also been progress in improving the ELBO estimates (Sinha & Dieng, 2021; Burda et al., 2016; Masrani et al., 2021) and image synthesis (Child, 2021; Vahdat & Kautz, 2021; Lee et al., 2020b; Xiao et al., 2021). Next, we compare our proposed approach in detail with several of these related existing model families.

**Unconditional DDPM**: DDPM/DDIM as introduced in (Ho et al., 2020; Song et al., 2021a) lacks a low-dimensional latent code which limits model application scope in several downstream tasks. In contrast, DiffuseVAE equips diffusion models with a low dimensional latent code that can be utilized for downstream tasks including but not limited to controllable synthesis. Moreover, we demonstrate a better speed vs quality tradeoff in DiffuseVAE as compared to standard unconditional DDPM/DDIM models and that the conditioning signal in DiffuseVAE helps in generalization to noisy conditioning signals.

**Conditional DDPM**: Conditional DDPM as introduced in (Ho et al., 2021) and (Saharia et al., 2021) uses a cascade of multiple diffusion models (CDMs) for generating high-resolution images. However, for even a two-stage pipeline, the sampling time of such models would be effectively much higher than DiffuseVAE. Given the flexibility of our approach, we hypothesize that a single-stage VAE can also be replaced by a complex multi-stage VAE architecture as proposed in (Child, 2021; Vahdat & Kautz, 2021) for comparable sample quality to cascaded diffusion models without affecting the sampling time significantly. Moreover, such cascades lack a low-dimensional latent code which might be a limiting factor for certain downstream applications. It is worth noting that, (Ho et al., 2021) use a conditioning augmentation scheme where the high-resolution image is generated by conditioning on a blurred/noisy low resolution image. In contrast, our model is already conditioned on a reconstruction generated by a VAE (which is inherently blurry) and in some sense resembles the heuristic employed in CDMs.

**VAE based methods** Hierarchical VAEs (Sønderby et al., 2016; Vahdat & Kautz, 2021; Child, 2021; Razavi et al., 2019) can suffer from posterior collapse and heuristics like gradient skipping and spectral normalization (Miyato et al., 2018) might be required to stabilize training. Moreover, these models require a large dimensionality of the latent codes to generate high-fidelity samples (Vahdat & Kautz, 2021; Razavi et al., 2019). In contrast, DiffuseVAE training does not suffer from such instabilities and provides access to a single latent code layer (with dimensionality comparable to GANs) to generate high-fidelity samples. Among other recent works, VAEBM (Xiao et al., 2021) uses EBMs (Du & Mordatch, 2020; Nijkamp et al., 2019) to refine VAE samples while LSGM (Vahdat & Kautz, 2021) perform score-based modeling in the latent space of a VAE backbone. However, both VAEBM and LSGM use NVAE (Vahdat & Kautz, 2021) as the base VAE architecture which also lacks a low-dimensional latent code. (Lee et al., 2020b) *distill* the disentanglement properties in the VAE latent code to the latent space of a GAN-based generator. However, this approach would also suffer from existing problems of training stability and mode-collapse in GAN-based models. On the other hand, DiffuseVAE does not suffer from such problems

# D   Detailed Proofs

## D.1   Derivation of the DiffuseVAE objective

Given a high-resolution image $x_0$, an auxiliary conditioning signal $y$ to be modelled using a VAE, a latent representation $z$ associated with $y$, and a sequence of $T$ representations $x_{1:T}$ learned by a diffusion model, the DiffuseVAE generative process, $p(x_{0:T}, y, z)$ can be factorized as follows:

$$p(x_{0:T}, y, z) = p(z)p_\theta(y|z)p_\phi(x_{0:T}|y, z) \tag{40}$$

where $\theta$ and $\phi$ are the parameters of the VAE decoder and the reverse process of the conditional diffusion model, respectively. The log-likelihood of the training data can then be obtained as:

$$\log p(x_0, y) = \log \int p(x_{0:T}, y, z) dx_{1:T} dz \tag{41}$$

Furthermore, since the joint posterior $p(x_{1:T}, z|y, x_0)$ is intractable to compute, we approximate it using a surrogate posterior $q(x_{1:T}, z|y, x_0)$ which can also be factorized into the following conditional distributions:

$$q(x_{1:T}, z|y, x_0) = q_\psi(z|y, x_0)q(x_{1:T}|y, z, x_0) \tag{42}$$

where $\psi$ are the parameters of the VAE recognition network ($q_\psi(z|y, x_0)$). Since computation of the likelihood in Eq. (41) is intractable, we can approximate it by computing a lower bound (ELBO) with respect to the joint posterior over the unknowns ($x_{1:T}$, $z$) as:

$$\log p(x_0, y) \geq \mathbb{E}_{q(x_{1:T}, z|x_0, y)} \left[ \log \frac{p(x_{0:T}, y, z)}{q(x_{1:T}, z|x_0, y)} \right] \tag{43}$$

Plugging the factorial forms of the DiffuseVAE generative process and the joint posterior defined above in eqn. (43), we can simplify the ELBO as follows:

$$\log p(x_0, y) \geq \mathbb{E}_{q(x_{1:T}, z|y, x_0)} \left[ \log \frac{p(x_{0:T}, y, z)}{q(x_{1:T}, z|y, x_0)} \right] \tag{44}$$

$$\geq \mathbb{E}_{q(x_{1:T}, z|x_0, y)} \left[ \log \frac{p(z)p_\theta(y|z)p_\phi(x_{0:T}|y, z)}{q_\psi(z|y, x_0)q(x_{1:T}|y, z, x_0)} \right] \tag{45}$$

$$\geq \mathbb{E}_{q(x_{1:T}, z|x_0, y)} \left[ \log \frac{p(z)}{q_\psi(z|y, x_0)} + \log p_\theta(y|z) + \log \frac{p_\phi(x_{0:T}|y, z)}{q(x_{1:T}|y, z, x_0)} \right] \tag{46}$$

$$\geq \mathbb{E}_{q(z|y, x_0)} \left[ \log \frac{p(z)}{q_\psi(z|y, x_0)} + \log p_\theta(y|z) \right] + \mathbb{E}_{q(x_{1:T}, z|x_0, y)} \left[ \log \frac{p_\phi(x_{0:T}|y, z)}{q(x_{1:T}|y, z, x_0)} \right] \tag{47}$$

$$\geq \underbrace{\mathbb{E}_{q_\psi(z|y, x_0)} \left[ p_\theta(y|z) \right] - \mathcal{D}_{KL}(q_\psi(z|y, x_0)||p(z))}_{\mathcal{L}_{\text{VAE}}} + \mathbb{E}_{z \sim q(z|y, x_0)} \left[ \underbrace{\mathbb{E}_{q(x_{1:T}|y, z, x_0)} \left[ \frac{p_\phi(x_{0:T}|y, z)}{q(x_{1:T}|y, z, x_0)} \right]}_{\mathcal{L}_{\text{DDPM}}} \right] \tag{48}$$

## D.2   Derivation of the DiffuseVAE (Formulation-2) marginals

Given:

$$q(x_1|x_0, \hat{x}_0) = \mathcal{N}(\sqrt{1 - \beta_1}x_0 + \hat{x}_0, \beta_1 I) \tag{49}$$

$$q(x_t|x_{t-1}, \hat{x}_0) = \mathcal{N}(\sqrt{1 - \beta_t}x_{t-1} + (1 - \sqrt{1 - \beta_t})\hat{x}_0, \beta_t I) \tag{50}$$

From Eqn.(50), we can write,

$$x_t = \sqrt{1 - \beta_t}x_{t-1} + (1 - \sqrt{1 - \beta_t})\hat{x}_0 + \epsilon\sqrt{\beta_t}, \quad \text{where} \quad \epsilon \sim \mathcal{N}(0, I) \tag{51}$$

Taking expectations both sides,

$$\mathbb{E}(x_t) = \sqrt{1 - \beta_t}\mathbb{E}(x_{t-1}) + (1 - \sqrt{1 - \beta_t})\hat{x}_0 \tag{52}$$

$$\mathbb{E}(x_t) = \sqrt{1 - \beta_t}\left[\sqrt{1 - \beta_{t-1}}\mathbb{E}(x_{t-2}) + (1 - \sqrt{1 - \beta_{t-1}})\hat{x}_0\right] + (1 - \sqrt{1 - \beta_t})\hat{x}_0$$

$$\mathbb{E}(x_t) = \sqrt{(1 - \beta_t)(1 - \beta_{t-1})}\mathbb{E}(x_{t-2}) + \left(1 - \sqrt{(1 - \beta_t)(1 - \beta_{t-1})}\right)\hat{x}_0$$

$$\vdots \tag{53}$$

$$\mathbb{E}(x_t) = \sqrt{\prod_{t=2}^{t}(1 - \beta_t)}\mathbb{E}(x_1) + \hat{x}_0\left(1 - \sqrt{\prod_{t=2}^{t}(1 - \beta_t)}\right) \tag{54}$$

Substituting $\mathbb{E}(x_1) = \sqrt{1 - \beta_1}x_0 + \hat{x}_0$ from Eqn.(49) into the above formulation we get,

$$\mathbb{E}(x_t) = \sqrt{\prod_{t=1}^{t}(1 - \beta_t)}x_0 + \hat{x}_0 = \sqrt{\bar{\alpha}_t}x_0 + \hat{x}_0 \tag{55}$$

Similarly it can be shown that $Var(x_t) = (1 - \bar{\alpha}_t)I$. Therefore,

$$q(x_t|x_0, \hat{x}_0) = \mathcal{N}(\sqrt{\bar{\alpha}_t}x_0 + \hat{x}_0, (1 - \bar{\alpha}_t)I) \tag{56}$$

| Method | FID@10k $\downarrow$ |
|---|---|
| DiffuseVAE ($\hat{x}_0$) | **5.94** |
| DiffuseVAE ($\hat{x}_0$ + Latent code) | 6.07 |

| Method | FID@10k $\downarrow$ |
|---|---|
| DiffuseVAE (Two-stage) | **6.81** |
| DiffuseVAE (End-to-end) | 8.12 |

Table 7: FID (10k samples) comparison between different DiffuseVAE conditioning schemes on CIFAR10.

Table 8: FID (10k samples) comparison between two-stage and end-to-end training on CIFAR10.

## E  Justification of the design choices in DiffuseVAE

Here we justify the design choices made in the DiffuseVAE model specification.

1. **Choice of the conditioning signal** $y$: The choice of assuming the conditioning signal $y$ in Eq. 10 to be the training data $x_0$ is motivated by the task of *refining* the blurry samples generated by a simple VAE model using a DDPM model.

2. **Choice of the conditioning signal** $z$: The choice of conditioning the DDPM model on $\hat{x}_0$ (the VAE reconstruction of the training data $x_0$) instead of the VAE inferred latent code $z$ (usually lower-dimensional) allows us to condition the second stage DDPM directly on samples drawn from another model (not necessarily VAE) or on real images, which can be quite useful as illustrated in Section 4.5. Additionally, there can be a variant of our method in which the DDPM model is conditioned on both $z$ and $\hat{x}$. We conditioned the DDPM decoder on $z$ using Adaptive group normalization layers (Dhariwal & Nichol, 2021; Wu & He, 2018) as follows:

$$y = \texttt{MLP}(z) + e_t \tag{57}$$

$$\texttt{AdaGN}(h, y) = y_s \texttt{GroupNorm}(h) + y_b \tag{58}$$

where $h$ is the output of the first convolution in the residual block and $y = [y_s, y_b]$ is obtained from the latent code $z$ and the time-step embedding $e_t$. On benchmarking this DiffuseVAE (Formulation-1) variant on CIFAR-10 trained for around 1.1M steps, we found that the resulting model exhibited slightly worse performance compared to the DiffuseVAE variant conditioned only on the VAE reconstructions ($\hat{x}_0$) (See Table 7). Therefore, we only condition the DDPM model in DiffuseVAE only on the VAE generated reconstructions.

3. **Two-stage training**: The choice of a two-stage training approach in DiffuseVAE is motivated by two reasons. Firstly, in our early experiments on CIFAR-10, we observed that the end-to-end model exhibited much worse performance than its two-stage counterpart during inference (See Table 8) where both models were trained for 400k steps. Secondly, from a computational standpoint, using a two-stage training formulation would be more amenable to training on limited compute resources as end-to-end training would require both models to fit in memory.

|  |  | CIFAR-10 | CelebA-64 | CelebA-HQ-128 | CelebA-HQ-256 | LHQ-256 |
|---|---|---|---|---|---|---|
| | **Stage-I VAE Hyperparameters** | | | | | |
| **Data** | Resolution | 32 x 32 | 64 x 64 | 128 x 128 | 256 x 256 | 128 x 128 |
| | Data Range | [0, 1] | [0, 1] | [0, 1] | [0, 1] | [0, 1] |
| **Model** | Architecture | See Code | See Code | See Code | See Code | See Code |
| | # of parameters | 9.2M | 14M | 21.1M | 32.7M | 36.3M |
| **Training** | Random Seed | 0 | 0 | 0 | 0 | 0 |
| | Mixed Precision | No | No | No | No | No |
| | Effective Batch Size | 128 | 128 | 128 | 32 | 256 |
| | # of epochs | 500 | 250 | 500 | 500 | 500 |
| | Optimizer | Adam(lr=1e-4) | Adam(lr=1e-4) | Adam(lr=1e-4) | Adam(lr=1e-4) | Adam(lr=1e-4) |
| | Latent code size | 512 | 512 | 1024 | 1024 | 1024 |
| | Ex-PDE | GMM(N=50) | GMM(N=75) | GMM(N=100) | GMM(N=100) | GMM(N=100) |
| | KL-weight | 1.0 | 1.0 | 1.0 | 1.0 | 1.0 |
| | **Stage-II DDPM Hyperparameters** | | | | | |
| **Data** | Resolution | 32 x 32 | 64 x 64 | 128 x 128 | 256 x 256 | 256 x 256 |
| | Horizontal Flip | Yes | Yes | Yes | Yes | Yes |
| | Data Range | [-1, 1] | [-1, 1] | [-1, 1] | [-1, 1] | [-1, 1] |
| **Model** | # of channels | 128 | 128 | 128 | 128 | 128 |
| | Scale(s) of attention block | [16] | [16] | [16] | [16] | [16,8] |
| | # of attention heads | 8 | 8 | 8 | 8 | 8 |
| | # of residual blocks per scale | 2 | 2 | 2 | 2 | 2 |
| | Channel multipliers | (1,2,2,2) | (1,2,2,2,4) | (1,2,2,3,4) | (1,1,2,2,4,4) | (1,1,2,2,4,4) |
| | # of parameters | 35.7M | 84.6M | 95.2M | 113M | 114M |
| | Dropout | 0.3 | 0.1 | 0.1 | 0.1 | 0.1 |
| | Noise Schedule (default) | Linear(1e-4, 0.02) | Linear(1e-4, 0.02) | Linear(1e-4, 0.02) | Linear(1e-4, 0.02) | Linear(1e-4, 0.02) |
| | # of time-steps (T) | 1000 | 1000 | 1000 | 1000 | 1000 |
| **Training** | Random seed | 0 | 0 | 0 | 0 | 0 |
| | Mixed Precision | No | No | No | No | No |
| | EMA decay rate | 0.9999 | 0.9999 | 0.9999 | 0.9999 | 0.9999 |
| | Effective batch size | 128 | 128 | 64 | 64 | 64 |
| | # of steps | 1.1M | 0.54M | 0.46M | 0.36M | 0.35M |
| | Optimizer | Adam(lr=2e-4) | Adam(lr=2e-4) | Adam(lr=2e-5) | Adam(lr=2e-5) | Adam(lr=2e-5) |
| | Grad. Clip Threshold | 1.0 | 1.0 | 1.0 | 1.0 | 1.0 |
| | # of lr annealing steps | 5000 | 5000 | 5000 | 5000 | 5000 |
| | Diffusion loss type | Noise prediction (L2) | Noise prediction (L2) | Noise prediction (L2) | Noise prediction (L2) | Noise prediction (L2) |
| **Evaluation** | Variance | fixedlarge | fixedlarge | fixedsmall | fixedsmall | fixedsmall |

Table 9: Hyperparameters for the training setup in DiffuseVAE

## F   Training and Hyperparameter details

All hyperparameters details related to VAE and DDPM training in DiffuseVAE are listed in Table 9. Moreover, all hyperparameters (model and training) were shared between both DiffuseVAE formulations.

**Data preprocessing**: During the first stage VAE training, all training data was normalized between [0.0, 1.0]. For the second stage DDPM training, the training data was scaled between [-1.0, 1.0] (including unconditional baselines and DiffuseVAE formulations). We also applied random horizontal flips as a form of data augmentation to the training images during the second stage DDPM training

**Model architecture**: We use the same network architectures as explored in prior work in diffusion models (Ho et al., 2020; Dhariwal & Nichol, 2021; Nichol & Dhariwal, 2021). The VAE architecture used for Stage-1 training consists of residual block architectures inspired from (Child, 2021) (Refer to our code for exact architectural details). The VAE latent code size was set to 1024 for LHQ-256 and CelebA-HQ (both 128 and 256 resolution variants) and 512 for the CIFAR-10 and CelebA (64 x 64) datasets. We do not investigate the effect of the size of the latent code in this work. Similar to prior work (Ho et al., 2020), for all datasets except CIFAR-10 models used in SoTA comparisons, we use the U-Net (Ronneberger et al., 2015) decoder implementation from (Nichol & Dhariwal, 2021) in the reverse process in Stage-II DDPM training. For the CIFAR-10 dataset, we used the U-Net decoder implementation from DDIM (Song et al., 2021a) (https://github.com/ermongroup/ddim/blob/main/models/diffusion.py). The U-Net decoder model hyperparameters are listed in Table 9.

**Training and Inference**: Unless specified otherwise, we use the same hyperparameters during training as proposed in (Ho et al., 2020). All DDPM models were trained using the simplified objective proposed in (Ho et al., 2020). We used a mix of 4 Nvidia 1080Ti GPUs (44GB memory), a cloud TPUv2-8 (64GB memory) and a cloud TPUv3-8 (128GB memory) for training the models. Specifically, we used the GPU setup for training our CIFAR-10 and CelebA-64 models while we utilized the TPUv2-8 for training CelebA-HQ

models at the 128 x 128 resolutions. Finally, we utilized the TPUv3-8 model for training on CelebA-HQ and LHQ models at 256 x 256 resolution.

**Evaluation**: For FID (Heusel et al., 2018) score computation, we utilized 10k samples for the CelebA-HQ-128 dataset and 50k samples for state-of-the-art comparisons on the CIFAR-10 and the CelebA-64 datasets. For CelebA-HQ 256 comparisons we computed FID scores on 30k samples since the CelebA-HQ dataset contains 30k images. We used the `torch-fidelity` (Obukhov et al., 2020) package for FID and IS score computations. In this work, when saving samples to disk, we used standard denormalization (i.e. $0.5 * \mathrm{img} + 0.5$) for all datasets. We used our GPU setup primarily for evaluation.

| | CIFAR-10 (FID@50k) | CelebA-64 (FID@50k) | CelebA-HQ-256 (FID@10k) |
|---|---|---|---|
| Baseline VAE | 137.68 | 72.11 | 97.07 |
| DiffuseVAE (Form-1, Ex-PDE) | **2.83** | **4.05** | **11.28** |

Table 10: Quantitative comparison between sample quality of first stage VAEs in DiffuseVAE (Generator) and the final DiffuseVAE samples (Refiner)

| | CelebA-64 | | | | CIFAR-10 | | | |
|---|---|---|---|---|---|---|---|---|
| | 10 | 25 | 50 | 100 | 10 | 25 | 50 | 100 |
| DDPM (uncond) | 37.31 | 17.06 | 10.99 | 8.26 | 42.66 | **15.97** | **9.98** | **7.76** |
| DiffuseVAE (Form-1, Ex-PDE) | 26.09 | 14.16 | 9.58 | 7.54 | **34.19** | 16.74 | 11.00 | 8.48 |
| DiffuseVAE (Form-2, Ex-PDE) | **25.79** | **13.89** | **9.09** | **7.15** | 34.22 | 17.36 | 11.00 | 8.28 |

Table 11: Speed vs quality tradeoff comparison between DDPM and DiffuseVAE for the CIFAR-10 and CelebA-64 datasets. FID reported using 10k samples

## G  Additional Results

### G.1  Generator-Refiner Framework

Some additional qualitative results demonstrating the generator-refiner framework in VAEs are shown in Fig. 13. Table 10 further supports our qualitative results for several other benchmarks by comparing the FID scores between Stage-1 VAE generated samples and the corresponding final DiffuseVAE samples.

### G.2  Controllable synthesis

The directions for meaningful concepts (or image attributes like gender, age, hair style) are obtained by considering pairs of attribute negative and positive training samples. For each such pair, we compute the latent code representation for the positive and the negative sample and compute the difference between the attribute positive and the negative latent. We repeat this procedure for all such pairs and compute the average of the difference between the latent codes to obtain the direction vector for the attribute. Formally, given an attribute of interest $a$ and the a set of tuples $(x_{pos}^{(i)}, x_{neg}^{(i)})_{i=1}^{N}$ of attribute positive and negative images, the latent direction $z_a$ is given by:

$$z_a = \frac{1}{N} \sum_{i=1}^{N} \left[ f(x_{pos}^{(i)}) - f(x_{neg}^{(i)}) \right] \tag{59}$$

where $f$ denotes a mapping from the image to the latent space (the VAE encoder in this case). Given this latent direction, we can manipulate an attribute negative image by simply adding this vector to the latent code representation of the attribute negative image and decoding the resulting latent code representation as follows:

$$z_p = z_n + \lambda z_a \tag{60}$$

where $z_n$ is the latent code representation of the atribute negative image, $z_p$ is the new latent code containing the missing attribute and $\lambda$ is a scalar which controls the coarseness of the controllable generation (higher values usually result in more coarse generations). In this work, we use the attribute annotations provided by the CelebAMask-HQ dataset (Lee et al., 2020a) and a value of N=100 to construct the set of positive and negative samples for any attribute of interest. Additional controllable synthesis (including single attribute manipulation and composite manipulations) results for the CelebA-HQ dataset at the 128 x 128 resolution are shown in Fig. 14. Figure 15 compares between composite edit-based samples generated from our first stage VAE and the corresponding refined samples generated from DiffuseVAE.

|  | 10 | 25 | 50 | 100 |
|---|---|---|---|---|
| DDIM (uncond) | 15.19 | 8.00 | 6.76 | 6.24 |
| DiffuseVAE (DDIM, Form1, Ex-PDE) | **11.79** | **7.44** | **6.51** | **6.14** |
| DiffuseVAE (DDIM, Form2, Ex-PDE) | 12.15 | 7.63 | 6.62 | 6.22 |

Table 12: Speed vs quality tradeoff comparison between DDIM and DiffuseVAE for the CIFAR-10 dataset. FID reported using 10k samples

|  | 10 | 25 | 50 | 100 |
|---|---|---|---|---|
| DiffuseVAE (DDIM, Form-1) | 26.07 | 19.75 | 18.90 | 18.85 |
| DiffuseVAE (DDIM, Form-1, GMM=100) | **24.09** | **17.47** | **16.65** | **16.63** |

Table 13: FID scores (on 10k samples) for DiffuseVAE (Form-1) using DDIM sampling for the CelebA-HQ-256 dataset

## G.3 Speed vs quality tradeoffs

**Reverse Process subsequence selection**: We use the *linear* and *quadratic* time-step selection as discussed in DDIM (Song et al., 2021a), when running the reverse process for only a subsample of the time-steps for efficient sampling. We call this *spaced* sampling. For benchmarking both DiffuseVAE and the baseline DDPM/DDIM models for the speed vs quality tradeoff, we selected the scheme which yielded lower FID values. Hence, we use the quadratic time-step schedule for all datasets when benchmarking DiffuseVAE while we used the quadratic schedule for the CIFAR-10 and the CelebA-64 datasets and linear schedule for the CelebA-HQ dataset when benchmarking the baseline DDPM/DDIM. There is also a possibility of using *truncated* sampling in which only the last $t$ time-steps are used for sampling. However, we found that the latter yielded inferior results than spaced sampling, so we do not report the FID scores for truncated sampling here.

**Additional results on speed vs quality tradeoff**: Table 11 shows a speed vs quality tradeoff comparison between DiffuseVAE (with Ex-PDE) and the DDPM baseline for the CIFAR-10 and the CelebA-64 benchmarks. Both methods use the *fixedsmall* variance type as discussed in (Ho et al., 2020). On the CelebA-64 dataset, DiffuseVAE again provides a much better speed vs quality tradeoff than a standard DDPM. However, on the CIFAR10 dataset, DiffuseVAE lags behind the standard DDPM (except at T=10) in terms of FID scores. This is surprising, since for T=1000, our DiffuseVAE model outperforms our baseline DDPM. However, when using DDIM sampling, DiffuseVAE outperforms the unconditional DDIM (See Table 12). For completeness, we also report the FID scores on 10k samples for our CelebA-HQ-256 DiffuseVAE (Form-1) model using DDIM sampling in Table 13

## G.4 State-of-the-art Comparisons

**Model size and Runtime comparison: LSGM and DiffuseVAE**: Here we compare model sizes between DiffuseVAE and LSGM (Vahdat et al., 2021). Table 14 compares the model sizes between the LSGM and DiffuseVAE models on the CIFAR-10 and the CelebA-HQ-256 benchmarks. LSGM utilizes an order of magnitude larger VAE backbones and denoising decoders in comparison to DiffuseVAE. When computing the LSGM model size, we compute the size of the best FID model (See https://github.com/NVlabs/LSGM). To examine the performance gains when using larger models, we trained a DiffuseVAE (Form-1) model with an unchanged VAE baseline but with a larger DDPM decoder with around 73M parameters on CIFAR-10. Indeed, when using a larger model, DiffuseVAE with Ex-PDE achieves a FID of **2.62** and a mean IS of **9.75** on CIFAR-10 which shows that our model can take advantage of larger model sizes as well.

We further benchmarked DiffuseVAE and LSGM CIFAR-10 models in terms of the wall-clock time and memory required for sample generation on a batch size of 64 samples on a single Nvidia 1080Ti GPU. In terms of memory consumption, the LSGM model consumes 5.1GB in comparison to around 2.00GB consumed by DiffuseVAE. This is to be expected due to a larger LSGM model size. Interestingly, LSGM

|                                | CIFAR-10 | CelebA-HQ-256 |
|--------------------------------|----------|---------------|
| LSGM (VAE backbone)            | 86.6M    | 50.9M         |
| LSGM (Denoising decoder)       | 375.6M   | 408.4M        |
| DiffuseVAE (VAE backbone)      | 9.2M     | 32.7M         |
| DiffuseVAE (Denoising decoder) | 35.7M    | 113.3M        |

Table 14: Model size comparison (in terms of the number of parameters) between DiffuseVAE and LSGM on the CIFAR-10 and CelebA-HQ-256 benchmark

(using 140 NFEs) only takes 67.03s to generate a batch of 64 samples as compared to around 103.13s required by DiffuseVAE (using 1000 NFEs). We hypothesize that this gain is primarily due to the efficacy of applying diffusion in the latent space in LSGM as compared to the pixel-space in DiffuseVAE. However, this design choice also prevents access to a compact latent space in LSGM.

### G.5    Temperature Sampling in DiffuseVAE

: We experiment with a temperature scaling technique where during the DDPM sampling stage in DiffuseVAE, we sample the initial DDPM latent $x_T$ from a base Gaussian distribution with standard deviation scaled by $\lambda$. This is a common technique utilized in prior works (Vahdat & Kautz, 2021; Kingma & Dhariwal, 2018) to tradeoff between sample quality and diversity. Interestingly, we found that for CelebA-HQ-256 dataset, samples generated during intermediate training stages (and even after convergence) suffer from color bleeding as shown in Fig. 10 (Top Row). We found that by applying temperature annealing in the second stage DDPM latents alleviates this problem (See Fig. 10(Bottom Row)). Therefore, we compute FID for state-of-the-art comparisons on this benchmark with a scaling factor of 0.8. We did not observe such color channel bleeding in samples of other benchmarks. In such cases, we observed that temperature scaling did not help and thus was not used to report FID scores.

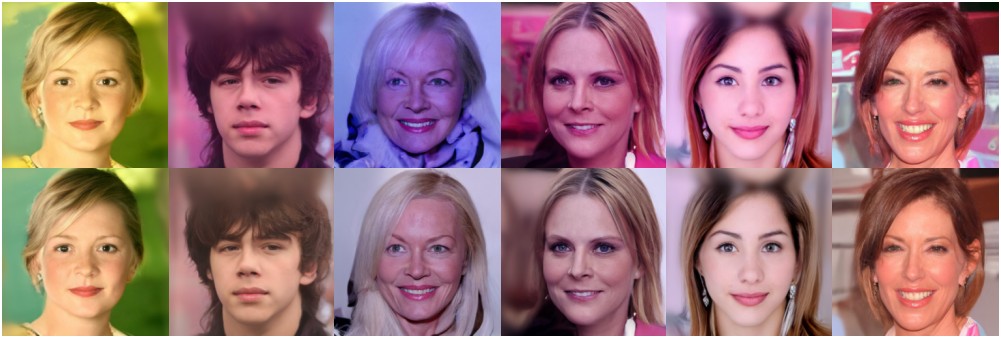

Figure 10: Effect of temperature sampling in DDPM latents in DiffuseVAE. (Top Row) Samples generated with $\lambda = 1.0$. (Bottom Row) Samples generated with $\lambda = 0.8$

### G.6    DiffuseVAE Training Dynamics and Stability

Although hierarchical VAEs (Vahdat & Kautz, 2021; Child, 2021) generate significantly better samples than a standard VAE (with a single stochastic layer) (Kingma & Welling, 2014), the former can be unstable to train and often require carefully designed heuristics like spectral normalization, gradient clipping etc. However, even with these heuristics, stable training is not guaranteed. In contrast, standard VAEs often do not suffer from training instability issues. Indeed our empirical results in Figure 11 suggest the same. Figure 11 shows VAE training dynamics during training for the CIFAR-10 (Top Row) and the CelebA-HQ 256 (Bottom Row) datasets. As expected, the reconstruction loss (Middle column) and the total loss (Right column) for both the datasets decrease as training progresses. On the other hand, the KL loss increases for both the datasets early during training (Left column). This can be expected since during training, the VAE posterior $q(z|x)$ becomes more complex so as to obtain a better reconstruction loss. Therefore, the divergence

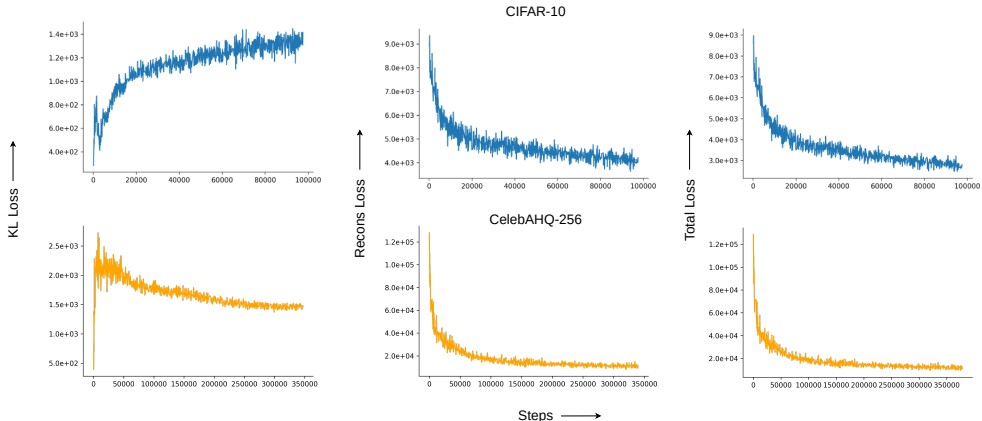

Figure 11: Illustration of VAE training dynamics on the CIFAR-10 (Top Row) and the CelebA-HQ 256 dataset (Bottom Row) datasets. The columns from left to right represent the variation in KL loss, Reconstruction Loss and Total Loss during training respectively.

between $q(z|x)$ and the prior $p(z)$ (in our case a standard gaussian) increases, leading to a higher KL Loss. Therefore, DiffuseVAE is more stable to train than the corresponding hierarchical VAE and GAN-based counterparts.

## G.7 Learning curve comparison between DiffuseVAE formulations

Figure 12 shows comparison between the learning curves between DiffuseVAE formulations for the CIFAR-10 and CelebA-HQ 128 benchmarks. We used this analysis to assess model convergence. For the CIFAR-10 dataset, our models started to slightly overfit after 2000 epochs, so we utilize the corresponding checkpoint for all analysis. For the CelebA-HQ 128 dataset, we stopped training after exhaustion of our maximum compute budget of 1000 epochs and utilize the corresponding checkpoint for subsequent analysis and comparisons.

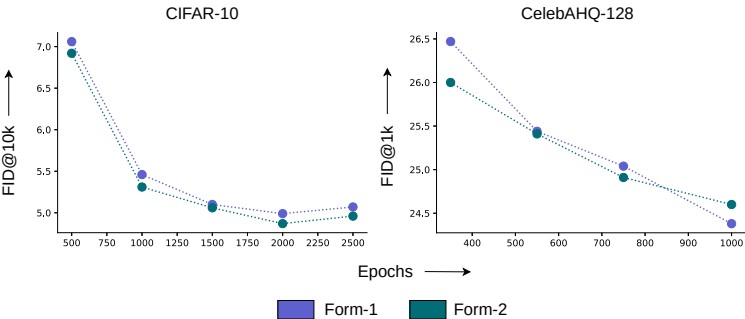

Figure 12: Learning curve (FID vs epochs) comparison between DiffuseVAE formulations for the CIFAR-10 (Left) and the CelebA-HQ-128 dataset (Right). T=1000 during inference

## G.8 Additional Samples

To demonstrate generalization to more complex scenes, some qualitative samples generated using a DiffuseVAE model trained on the LHQ-256 dataset with T=1000 and a temperature scaling factor of 0.8 are shown in Figure 17. Some additional samples from our CelebA-HQ model using DDIM sampling with 50 steps in the reverse process are shown in Figure 18. Figure 19 shows some additional samples from the same model with T=1000 and a temperature scaling factor of 0.8. Lastly, Figure 20 shows samples from the CelebAHQ-256 model but with shared latents in the DDPM stage (so effectively the generation is driven completely by low-dimensional latents from the VAE model).

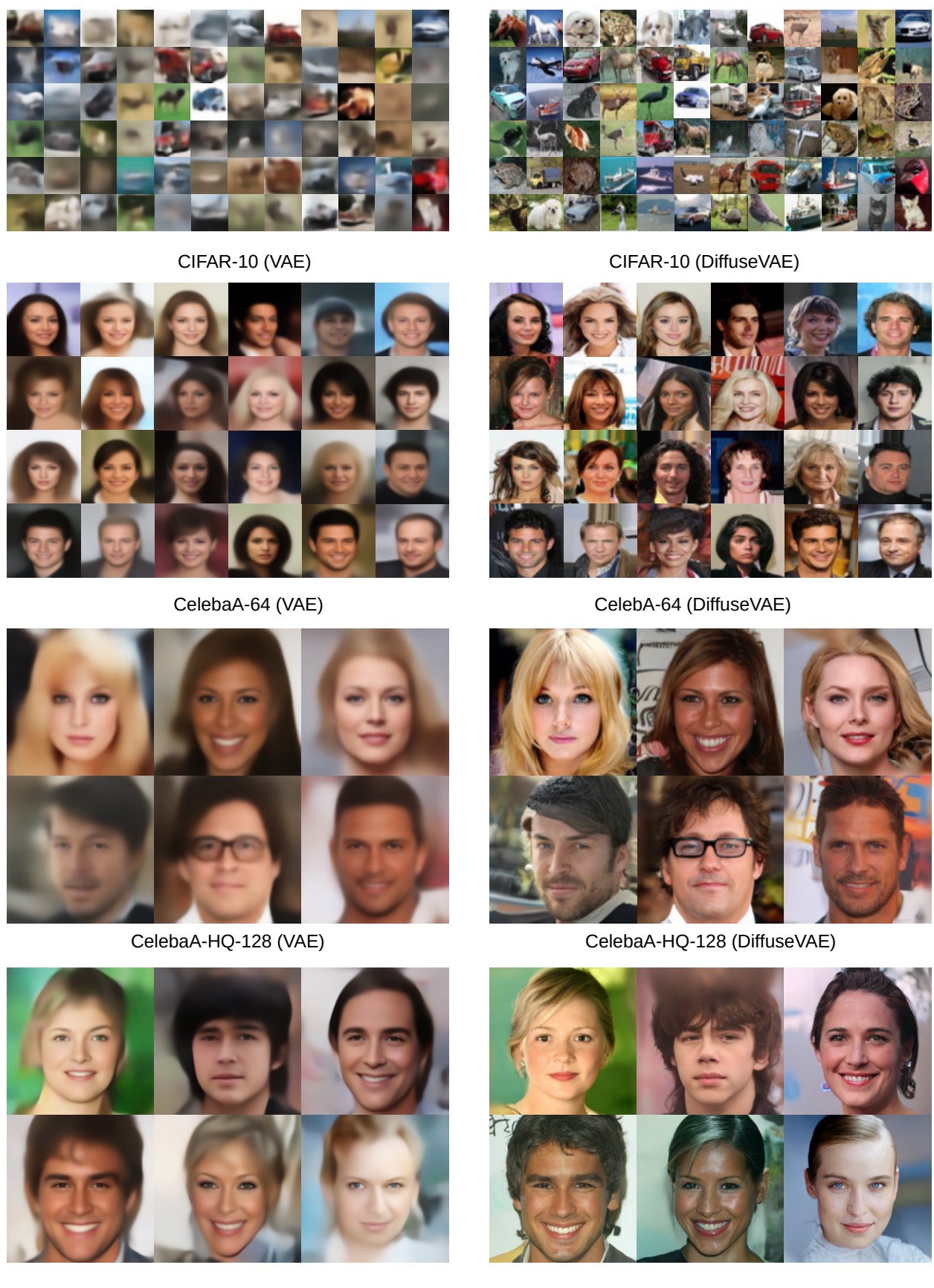

CIFAR-10 (VAE)                    CIFAR-10 (DiffuseVAE)

CelebaA-64 (VAE)                  CelebA-64 (DiffuseVAE)

CelebaA-HQ-128 (VAE)             CelebaA-HQ-128 (DiffuseVAE)

CelebaA-HQ-256 (VAE)             CelebaA-HQ-256 (DiffuseVAE)

Figure 13: Additional results demonstrating the generator-refiner framework in DiffuseVAE

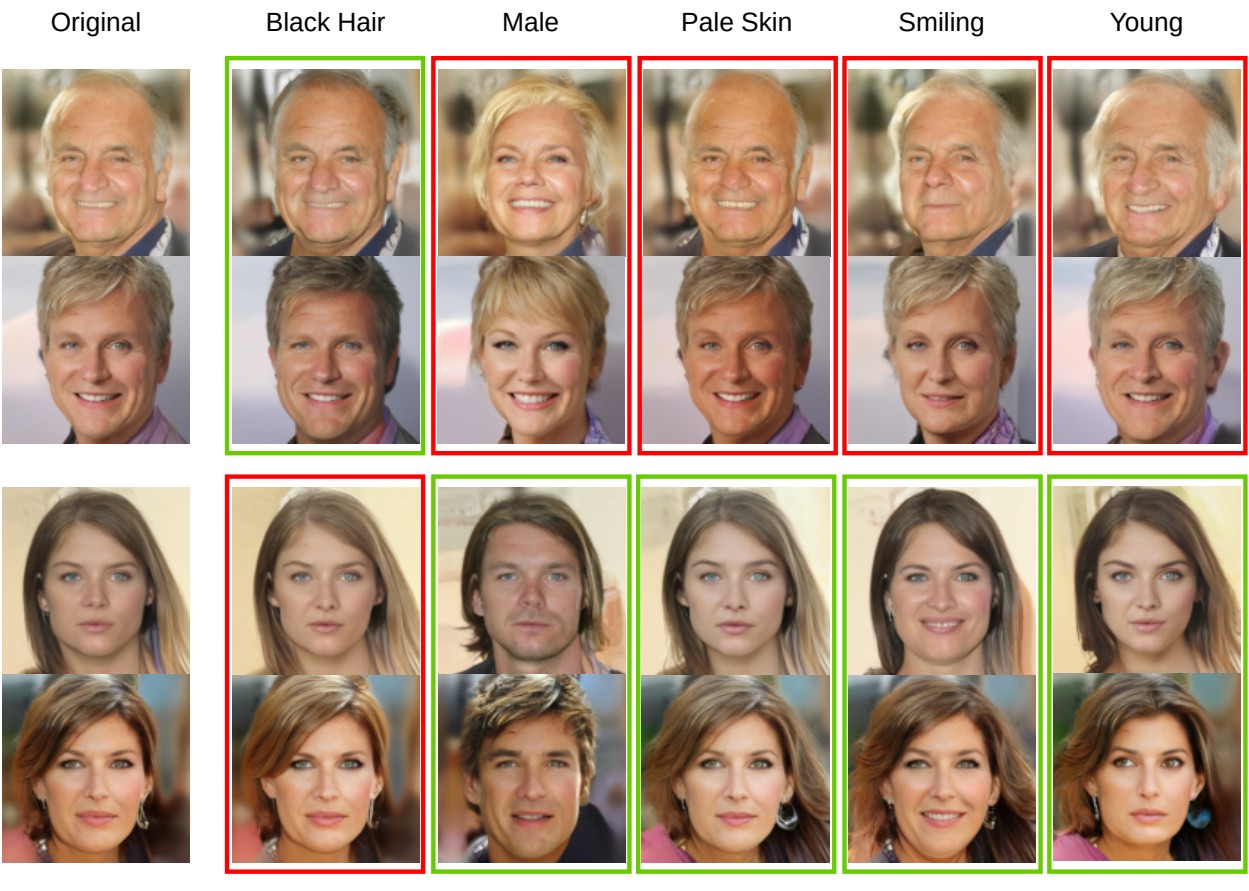

Composite Edits

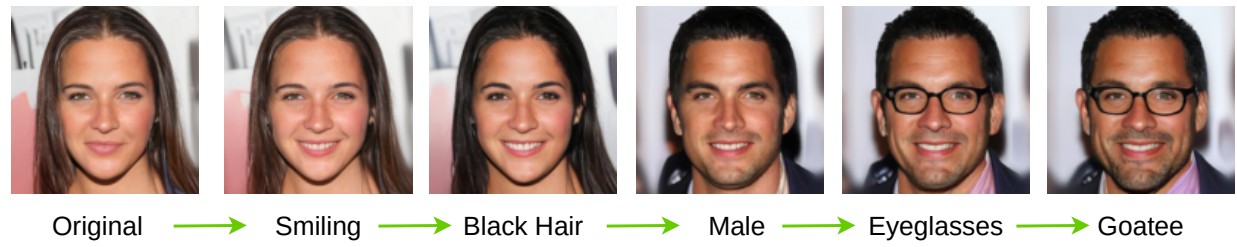

Original ⟶ Smiling ⟶ Black Hair ⟶ Male ⟶ Eyeglasses ⟶ Goatee

Figure 14: Additional results demonstrating controllable synthesis in the CelebA-HQ-128 dataset. Green boxes denote the vector addition operation while Red boxes denote the vector subtract operation

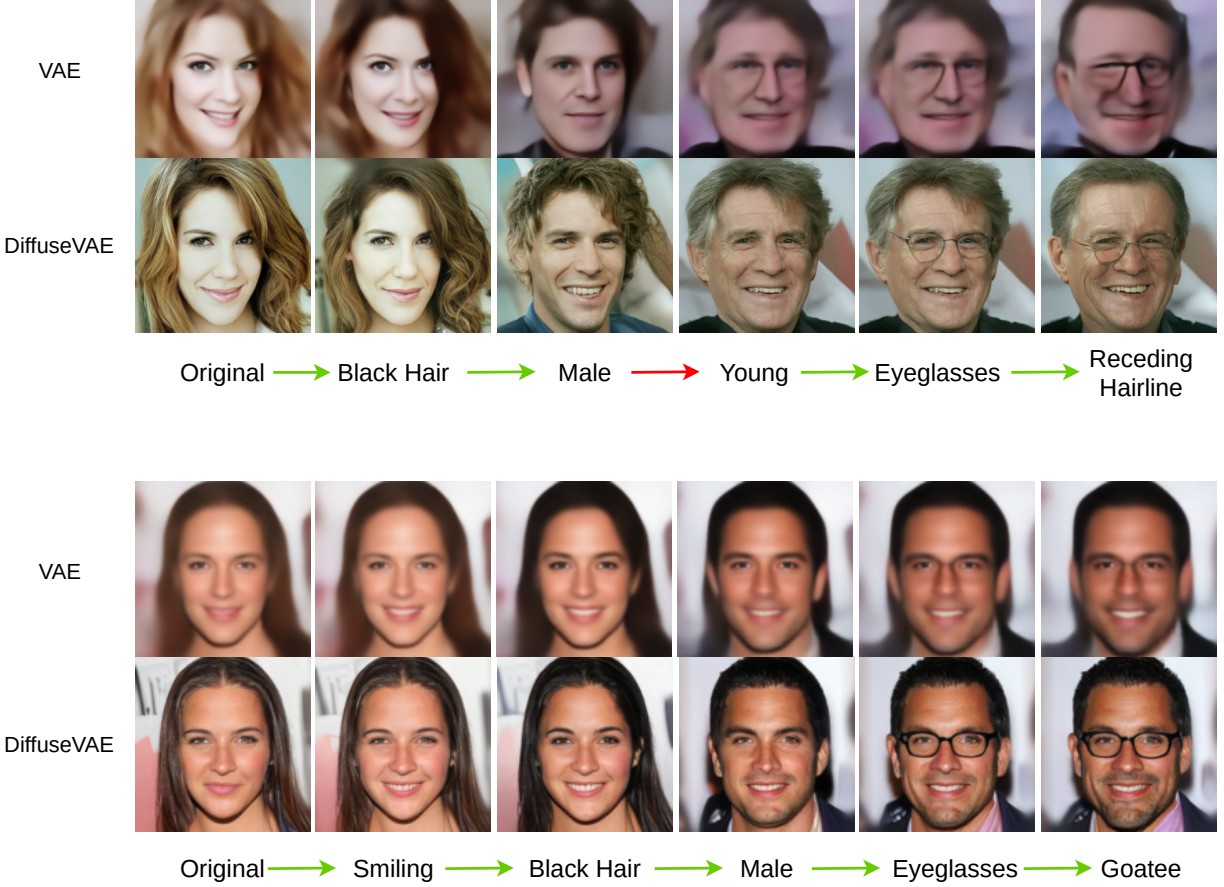

Figure 15: Comparison between composite edit samples generated using the first stage VAE vs the corresponding refined samples generated by DiffuseVAE.

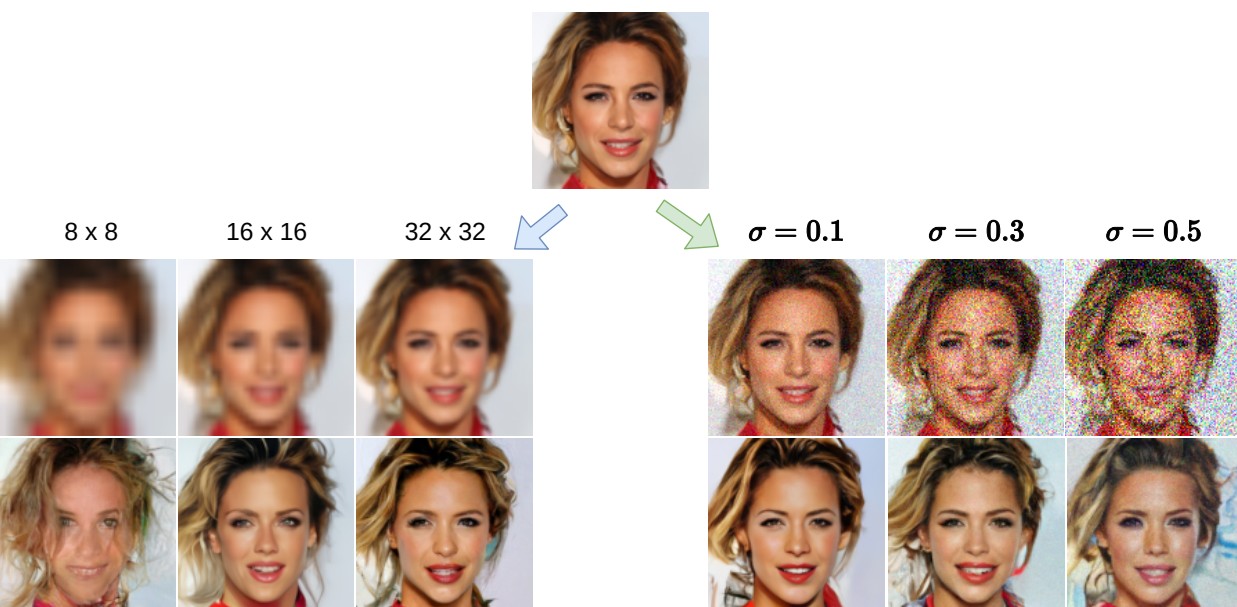

Figure 16: Illustration of DiffuseVAE generalization to different noise types in the conditioning signal on CelebA-HQ-128. $\sigma$ denotes the standard deviation of the gaussian noise added to the conditioning signal. As noise becomes more severe, the output generated by DiffuseVAE becomes significantly worse. All final samples generated using T=100 with DDIM sampling.

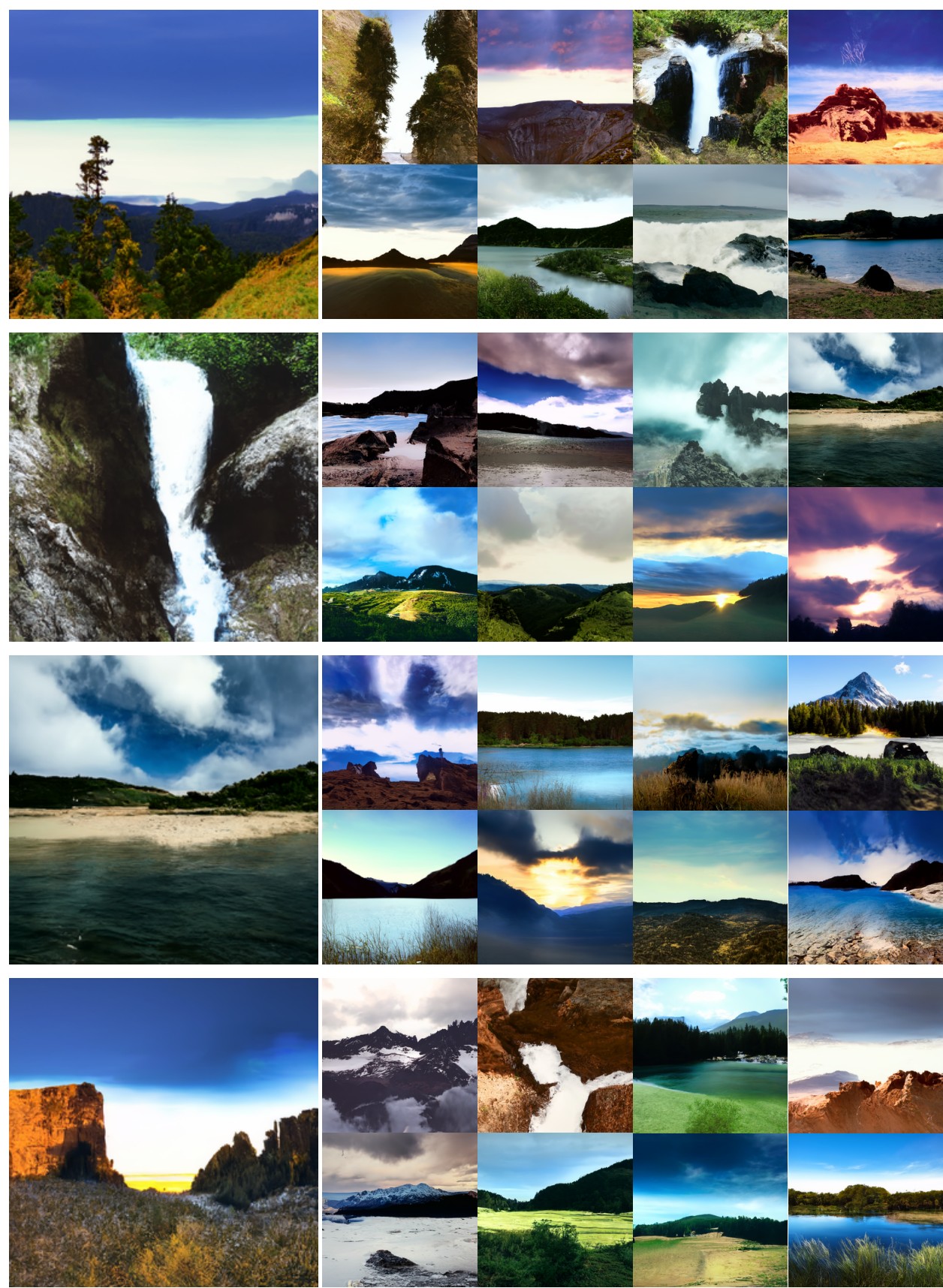

Figure 17: Samples generated from DiffuseVAE trained on the LHQ-256 dataset. T=1000 during sampling

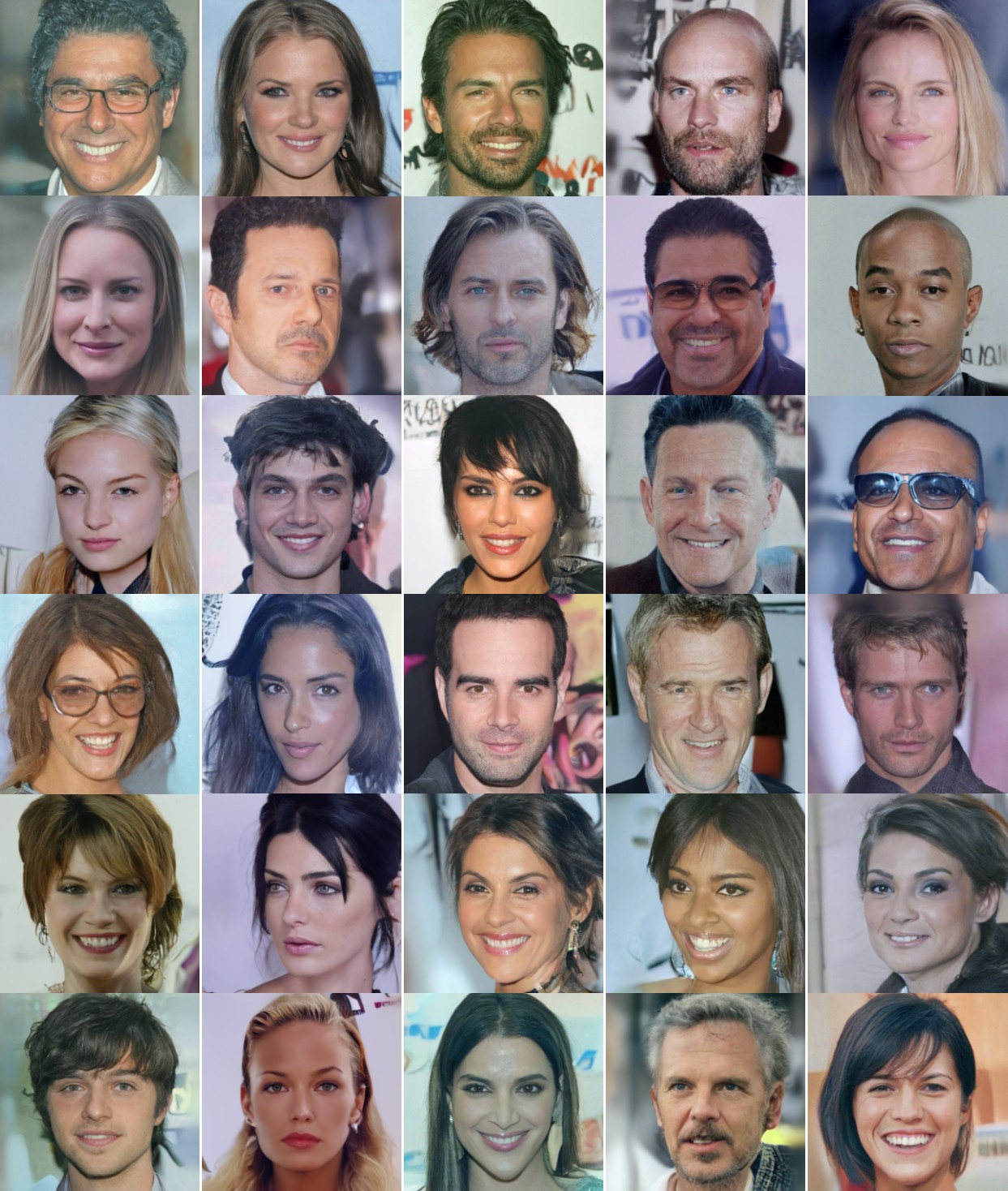

Figure 18: Additional samples from generated from DiffuseVAE trained on the CelebAHQ-256 dataset. T=50 using DDIM sampling.

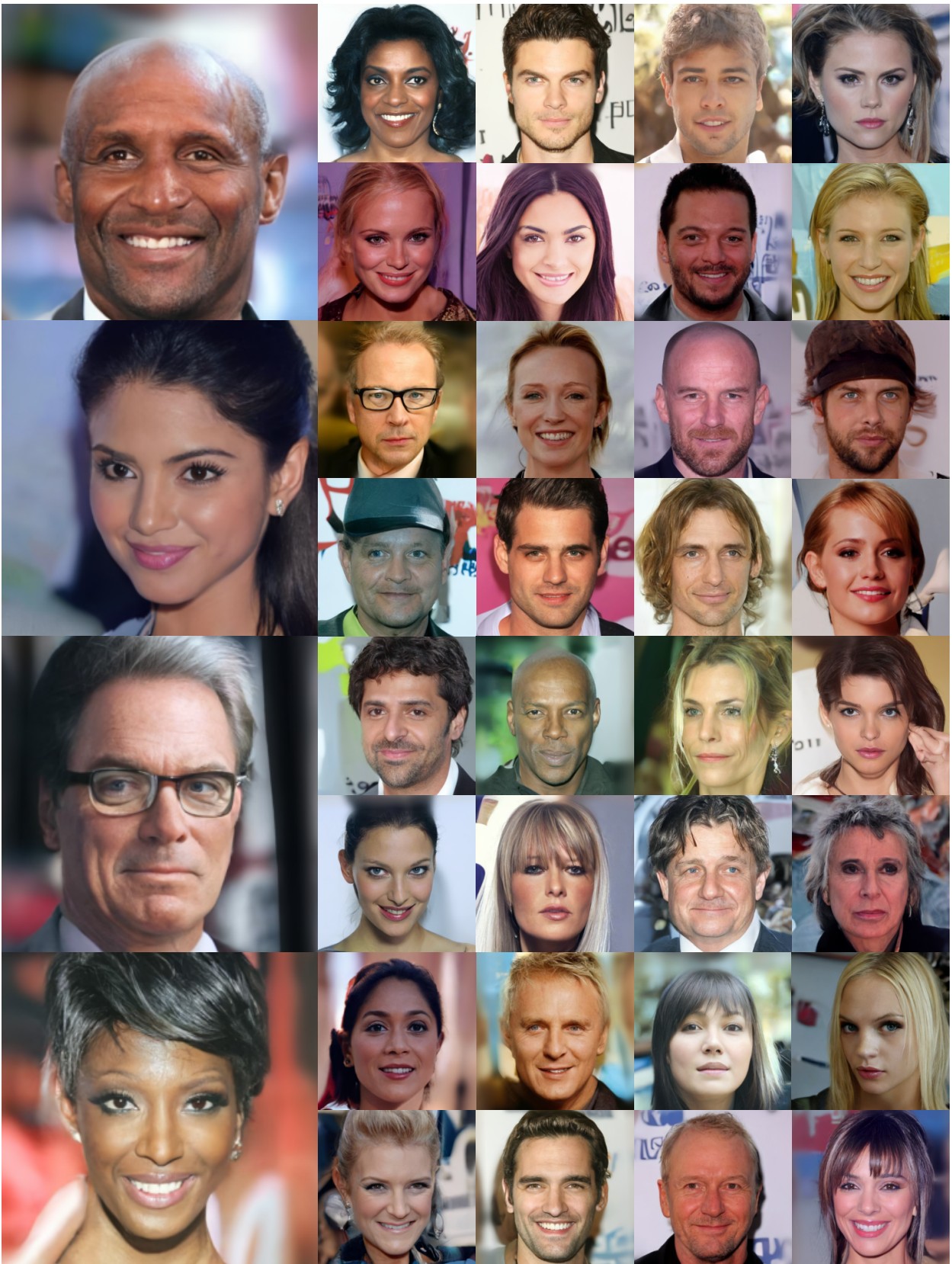

Figure 19: Additional samples from generated from DiffuseVAE trained on the CelebAHQ-256 dataset (T=1000, Temp. Scaling factor was set to 0.8 during sampling).

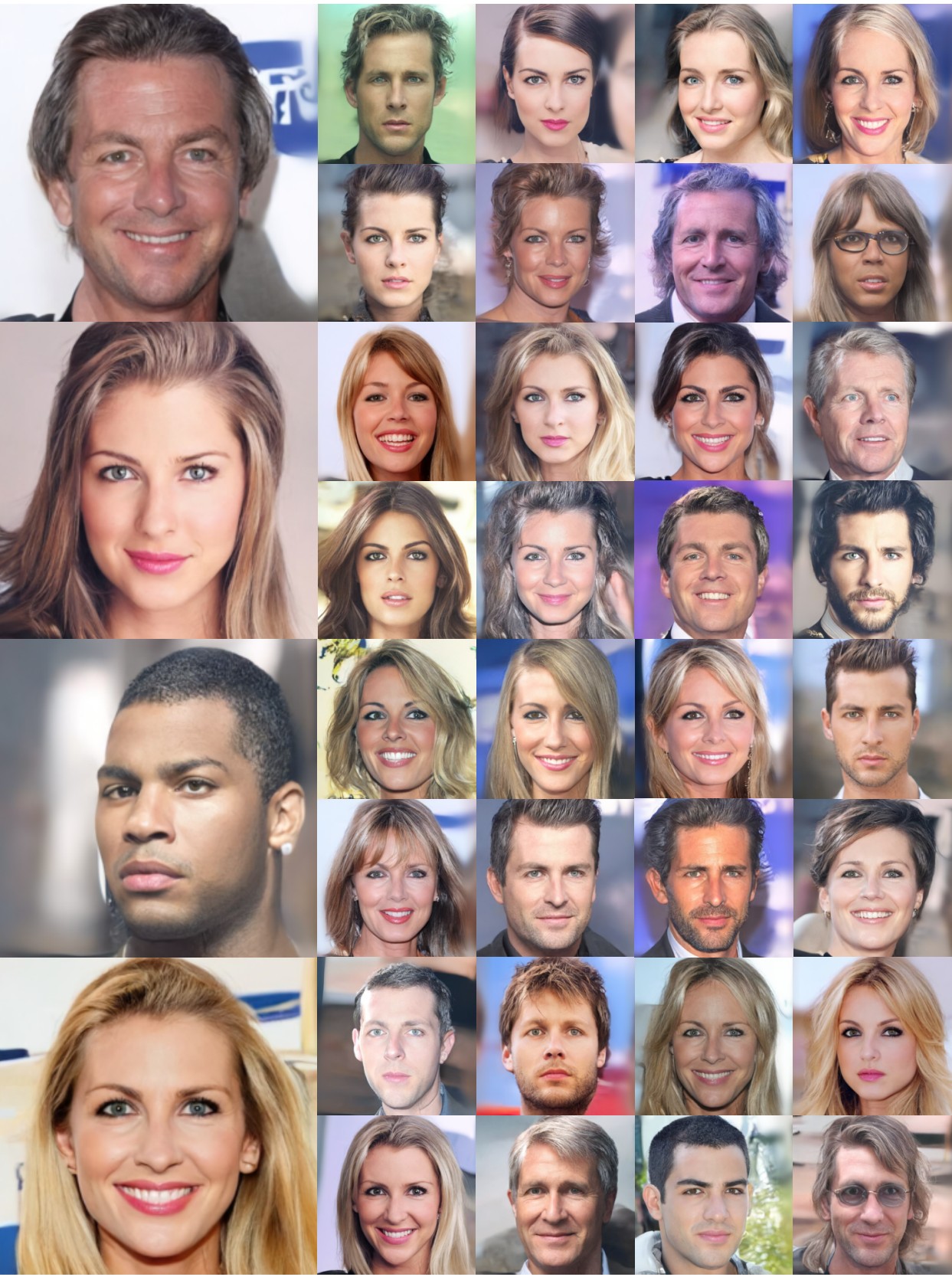

Figure 20: Additional samples from generated from DiffuseVAE trained on the CelebAHQ-256 dataset with the DDPM latents shared between samples. The generation is effectively driven by low-dimensional VAE latent space (T=1000, Temp. Scaling factor was set to 0.8 during sampling).

