# OpenReview forum: "DiffuseVAE: Efficient, Controllable and High-Fidelity Generation from Low-Dimensional Latents"
_TMLR — Accepted by TMLR_

### Review · Reviewer_5cpv · 2022-08-26

**Summary Of Contributions:**

This paper presents DiffuseVAE, which integrates VAE within a diffusion model, to design novel conditional parameterizations for diffusion models. In the experimental analyses, the proposed DiffuseVAE was examined in downstream tasks like controllable synthesis. The proposed method  improves the speed-quality tradeoff exhibited in standard unconditional DDPM/DDIM models. On  image synthesis benchmarks, the proposed model outperforms most existing VAE-based methods while performing on par with state-of-the-art models.

**Broader Impact Concerns:**

A Broader Impact Statement was given and some concerns were discussed.

**Requested Changes:**

- Training VAEs are challenging in general. Could you please elaborate training difficulties/challenges of the proposed DiffuseVAE and provide details regarding the employed tricks?

- Could you please provide learning curves for some of the analyses?

- DiffuseVAE is lighter than state-of-the-art by model size. Could you please also compare running times for inference and training of models?

- Qualitative results were given mainly for face images and low-resolution Cifar-10 images. Could you please provide additional results for higher resolution images containing other objects, such as for some samples from the Imagenet (even 128x128 are fine), scenes, etc. to evaluate generalization of the DiffuseVAE for larger class of objects.

**Strengths And Weaknesses:**

The strengths of the paper can be summarized as follows:

- The manuscript is well-written except a few minor problems in the statements. A clear review of the diffusion models was also given in the supplementary material.

- In the theoretical analyses, derivations of objections were given in detail.

- Some of the implementation details, such as network architectures and hyperparameters, were given.

- In the analyses, the DiffuseVAE outperforms VAE based methods.

One of the main weaknesses of the paper is the lower performance of DiffuseVAE compared to the state-of-the-art. Although DiffuseVAE boosts accuracy of VAEs and lighter than state-of-the-art higher performing models, its accuracy may need to be further improved to compete with the state-of-the-art.

---

> ### Author Response · Authors · 2022-09-21
> **Response to Reviewer 5cpv**
>
> We would like to thank the reviewer for their feedback on our work. We address the concerns from the reviewer as follows:
>
>
> 1.
> **Employed tricks:** We would like to point out that in this work we only consider standard Variational Autoencoders (as presented in [1]) which are not usually prone to training instabilities. Our empirical observations regarding VAE training suggest the same. This is in contrast to hierarchical VAEs like [2] which can require heuristics like spectral normalization, gradient clipping etc. for stable training. Therefore, we did not use any tricks for training the first stage VAE model in this work. We have added **Appendix G.6 **titled “**DiffuseVAE Training Dynamics and Stability**” to include plots of VAE learning dynamics on the CIFAR-10 and CelebA-HQ-256 dataset in our revision to illustrate the same.
>
>
> 2.
> **Learning curves:** In addition to the VAE training dynamics plots, we have included plots of FID vs the number of training steps for our CIFAR-10 and CelebA-HQ-128 models in the Appendix (**See Appendix G.7 titled “Learning curve comparison between DiffuseVAE formulations”**) as per the reviewers request.
>
>
> 3.
> **Training and Inference speed comparisons with SoTA**: Here we assume that the reviewer specifically refers to LSGM as SoTA, so we present comparisons with LSGM here. We benchmarked DiffuseVAE and LSGM CIFAR-10 models in terms of the wall-clock time and memory required for sample generation on a batch size of 64 samples on a single Nvidia 1080Ti GPU. In terms of memory consumption, the LSGM model consumes 5.1GB in comparison to around 2.00GB consumed by DiffuseVAE. This is to be expected due to a larger LSGM model size. Interestingly, LSGM (using 140 NFEs) only takes 67.03s to generate a batch of 64 samples as compared to around 103.13s required by DiffuseVAE (using 1000 NFEs). We hypothesize that this gain is primarily due to the efficacy of applying diffusion in the latent space in LSGM as compared to the pixel-space in DiffuseVAE. However, this design choice also prevents access to a compact latent space in LSGM. We have added the same in **Appendix G.4** of our revision. As per the reviewers request, we could not compare training times between DiffuseVAE and LSGM due to different compute resources used in both works and training our own LSGM model on CIFAR-10 was infeasible during the revision period.
>
>
> 4.
> **Additional Qualitative results:** We acknowledge the reviewer’s concerns about DiffuseVAE working on diverse, higher resolution datasets like ImageNet. However, running a large scale study like ImageNet with diffusion models requires significant compute requirements to be completed within a limited time frame. For instance, one state of the art ImageNet model ADM [3] employs a 296M, 422M and 554M parameter model for resolutions 64, 128 and  256 respectively on the ImageNet dataset. Training DiffuseVAE models with similar model sizes (for fair comparisons) would be computationally expensive for the compute budget available to us. However, accounting for reviewer concerns, we experimented with the LHQ [4] landscape dataset that consists of 90k high res landscape images (256 x 256) with diverse scenes. We have included these results in **Fig. 17** in our revision.
>
>
> 5.
> **Note on State-of-the-art:** One potential weakness raised by the reviewer is the gap between our model performance and SoTA. For the CIFAR-10 benchmark, our implementation of the baseline unconditional DDPM yields a FID score of 3.90, in contrast to 3.17 reported in [5] (See Table 4). During the response period, we found that this discrepancy is due to the underlying Unet baseline commonly used as a denoiser in diffusion models. After updating the Unet baseline from [6] to the one used in [7] (with approximately the same number of parameters for both Unet baselines), our unconditional DDPM now yields a FID score of 3.01 which is even better than the score reported in [5]. Consequently, our re-trained DiffuseVAE models (using the updated Unet baseline) with Formulation-1 and 2 achieve FID scores of 2.84 and 2.80 (with Ex-PDE) respectively. To further close the gap, a deeper DiffuseVAE model (with Form-1 and almost double the number of parameters) achieves a FID score of 2.60 with Ex-PDE. We would like to mention that the updated results on the CIFAR-10 benchmark are already highly competitive with recent SoTA models on this benchmark. We have updated **Table 4** in our revision to reflect the same. Since our CIFAR-10 results have been updated, the following sections from our previous paper version have also been updated in the revision: **Section 4.4**, **Tables 10, 11 and 12, Appendix G.4**

---

> > ### Author Response · Authors · 2022-09-21
> > **(Contd.)**
> >
> > **References:**
> >
> >
> > [1] Auto-Encoding Variational Bayes
> >
> > [2] NVAE: A Deep Hierarchical Variational Autoencoder
> >
> > [3] Diffusion Models Beat GANs on Image Synthesis
> >
> > [4] Aligning Latent and Image Spaces to Connect the Unconnectable
> >
> > [5] Denoising Diffusion Probabilistic Models
> >
> > [6] Improved Denoising Diffusion Probabilistic Models
> >
> > [7] Denoising Diffusion Implicit Models

---

### Review · Reviewer_4eBL · 2022-08-31

**Summary Of Contributions:**

This work proposed a generator-refiner framework, termed DiffuseVAE, that unifies the VAE and diffusion model for high-quality image generation. The DiffuseVAE training consists of two stages: 1) the standard VAE training that takes the original image as input and generates a reconstructed image, and 2) the diffusion model (i.e., DDPM) training by conditioning on the reconstructed images from VAE. Specifically, two formulations for the conditional diffusion model are proposed, where the first formulation only conditions the reverse denoising process on the VAE reconstruction (which has been commonly used in previous works), and the second formulation conditions both the forward diffusion and reverse denoising processes on the VAE reconstruction. Through extensive experiments on CIFAR-10, CelebA-64, and CelebA-HQ, this work demonstrates the effectiveness of the proposed method in generating high-quality samples, equipping DDPM with a low dimensional latent code  for controllable generation, and providing a better tradeoff between sample quality and number of steps.


**Broader Impact Concerns:**

I think the potential applications and the negative impacts have been well discussed in the paper.

**Requested Changes:**

Please see the weaknesses (which are critical to securing my recommendation) and minor concerns in the above.

**Strengths And Weaknesses:**

Strengths:
- The generator-refiner framework that uses a conditional diffusion model to refine the generated samples from VAE is an interesting idea. In particular, it combines the respective advantages of diffusion models and VAEs: diffusion models can generate high-quality images but they do not have a compact vector representations of images, while VAEs suffer from poor generation but they always have a fixed-length latent variable inferred from images for representation learning and attribute-based controllable generation.
- For the diffusion model training, two formulations of conditional DDPM are proposed. The idea of Formulation II that conditions both the forward and backward processes on the VAE output seems novel and very interesting.
- Experiments also showed that the generator-refiner framework can achieve better tradeoff between sample quality and number of steps. In particular, DiffuseVAE (with DDIM sampling) largely outperforms the standard unconditional DDIM for all $T \leq 100$.
- This work identifies the potential issue derived from the VAE training (i.e., the prior-hole problem) which may deteriorate the generation quality of DiffuseVAE, and proposes the post-fitting (with a GMM on the VAE latent space) method to improve the DiffuseVAE performance.

Weaknesses:
- This work starts from a more generalized case where the diffusion model conditions on a conditional signal $y$ and a latent representation $z$, but actually only considers the simplified setting for DiffuseVAE where $y=x_0$ and $z=\hat{x}_0$. Why not directly start from the simplified case, which is how DiffuseVAE is modeled and evaluated? I think it will make your presentation more focused and consistent.
- Although Formulation II for conditional DDPM is novel and interesting, its performance is not better than the previously used Formulation I across multiple experimental settings. This makes me wonder why we need Formulation II, which seems to be more complicated in training and sampling.
- For experiments on “state-of-the-art comparisons”, my major concern is whether DiffuseVAE can work well on more diverse, higher resolution datasets, such as ImageNet (res=64, 128 and 256). It would be more convincing to show comparable results with the latest state-of-the-art on ImageNet.
- For experiments on “controllable generation”, the results are also not quite good. In Figure 6, we can see that many image edits are not quite disentangled across different features. For example, the “black hair” editing is not sufficient, and the “male” editing is entangled with “mouth opening”, etc. Also, in Figure 6, it would be good to add the editing results of training VAE only to see where the potential failures/sufferings in image editing come from.
- For the experiments on generalizing to different noise types, I think the results (only on CIFAR-10 with some qualitative visualizations) are not quite convincing. First, in what conditions does your method perform well? For example, how large does the noise strength would be? What is the range for blurriness severity? Second, do we have similar results on CelebA-HQ?

Minor concerns:
- When “handling the DDPM stochasticity” by “sharing all stochasticity in the DDPM reverse process across all generated samples”, I wonder how to do it? Do you convert to a ODE-like deterministic sampling?
- In Table 4, what does “LSGM (FID)” mean? I didn’t see an explanation of how it differs from the standard LSGM.

---

> ### Author Response · Authors · 2022-09-21
> **Response to Reviewer 4eBL**
>
> We would like to thank the reviewer for their feedback on our work. We address the concerns stated in the Weaknesses section as follows:
>
> 1.
> **Justification for a generalized DiffuseVAE setting:** In this work, we present a novel framework which enables us to formally reason about learning the structure of conditioning signals and exploiting it in downstream generative models like diffusion models. A special case of this framework is DiffuseVAE in which reconstructions from a simple VAE are refined using a powerful diffusion model. However, the applications of this framework extend beyond DiffuseVAE. For instance, representing text using a latent space representation in the first stage can help in controllable synthesis during text-to-image generation applications. Infact, unCLIP [1] does just that and can be considered as a special case of a generic framework like ours in which the text representation is generated from a CLIP [2] encoder rather than a VAE encoder, while conditioning the downstream diffusion model on the text latents. Therefore, we believe that formalizing the generic case before discussing the specific DiffuseVAE model has merit. We also state our design decisions which lead to the DiffuseVAE model clearly in Section 3.2, thus adding a smooth transition from the general to the specific case. However, we are also open to moving the discussion of the generic framework after discussing the DiffuseVAE model formulation if the reviewers have a strong preference for it.
>
>
> 2.
> **Significance of Formulation-2:** Besides having a novel conditioning mechanism, we would like to refer the reviewer to Table 3 of the paper. In this case, combined with DDIM sampling, Formulation-2 provides better speed-quality tradeoff than Formulation-1 (For instance, DiffuseVAE Formulation-2 achieves a FID of **16.57 vs 18.01** achieved by Formulation-1 on the CelebA-HQ-128 benchmark when T=10 during DDIM sampling. See Table 3 for more details). More importantly, this improvement scales with the image resolution (See Tables 3 and 12). Interestingly, Formulation-2 (with Ex-PDE) has a FID score of 16.47 at T=10 for the CelebA-HQ-128 dataset which is better than the FID score obtained by an unconditional DDIM model at T=100, thus offering more than 10x speedup. We have made this empirical finding more explicit in our paper revision (using improved captioning in Table 3 and adding a note in Section 4.3).
>
>
> 3.
> **State of the art comparisons on ImageNet:** We acknowledge the reviewers concerns about DiffuseVAE working on diverse, higher resolution datasets like ImageNet. However, running a large scale study like ImageNet with diffusion models requires significant compute requirements to be completed within a limited time frame. For instance, one state of the art ImageNet model ADM [3] employs a 296M, 422M and 554M parameter model for resolutions 64, 128 and  256 respectively on the ImageNet dataset. Training DiffuseVAE models with similar model sizes (for fair comparisons) would be computationally expensive for the compute budget available to us. However, accounting for reviewer concerns, we experimented with the LHQ [4] landscape dataset that consists of 90k high res landscape images (256 x 256) with diverse scenes. We have included these results in **Figure. 17** in our revision.
>
>
> 4.
> **Controllable Synthesis results:** We thank the reviewer for raising this concern. We agree with including the corresponding VAE generated samples in the controllable synthesis results for improved debugging of some issues in controllability and have included a corresponding **Figure 15** in the Appendix (to avoid a larger main figure) and refer to the same in our concluding remarks in the paper (Section 6)
>
>
> 5.
> **Generalization to different noise types:** We agree with the reviewers comments regarding conditioning signal generalization and have included additional generalization results on CelebA-HQ samples with a wider range of noise std. (\sigma \in [0.1, 0.5]) and super-resolution settings in **Figure 16** of our revision. It is worth noting that, some artifacts in the generated samples are evident (which become more severe as the level of noise in the conditioning signal increases). This is discussed in **Section 4.5** in the revision.

---

> > ### Author Response · Authors · 2022-09-21
> > **(Contd.)**
> >
> > **Minor Concerns**
> >
> >
> >
> > * **Sharing DDPM stochasticity across samples:** We would like to point out that a single step in the reverse process involves sampling the timestep $x\_{t-1}$ from $x\_t$. This update looks as follows:
> > $
> > \begin{equation}
> > x_{t-1} = x_t + z \sigma_t
> > \end{equation}
> > $
> >
> >
> >     Therefore the DDPM sampling procedure has two sources of stochasticity: the initial latent $x_T$ sampled from an isotropic gaussian distribution and $z$ in the above eqn. which is also sampled from a standard gaussian. Due to a generator-refiner framework in DiffuseVAE, we propose to share these two sources of stochasticity between different model samples (_i.e.,_ freeze them to constants). Sharing this stochasticity or “style” leads to more consistent samples during controllable generation.
> >
> > * LSGM (FID): In the LSGM work, the authors train different models for better likelihoods and sample quality. LSGM (FID) refers to a setting when LSGM is optimized for best sample quality.
> >
> >
> >
> >
> > **References:**
> >
> >
> > [1] Hierarchical Text-Conditional Image Generation with CLIP Latents
> >
> > [2] Learning Transferable Visual Models From Natural Language Supervision
> >
> > [3] Diffusion Models Beat GANs on Image Synthesis
> >
> > [4] Aligning Latent and Image Spaces to Connect the Unconnectable

---

### Review · Reviewer_TFVU · 2022-09-07

**Summary Of Contributions:**

This paper proposes DiffuseVAE, a model that combines VAE and diffusion models.  DiffuseVAE is trained in a two-stage manner:
- Stage1: VAE Training: The VAE encoder takes the original image $x_0$ as input and generates a reconstruction $\hat{x}_0$
- Stage 2: Denoising Diffusion Probabilistic Models (DDPM) Training: $\hat{x}_0$ is used to condition the second stage DDPM.

Experiments are conducted on standard image synthesis benchmarks like CIFAR-10,  CelebA-64 and CelebA-HQ.  It shows that DiffuseVAE exhibits synthesis quality comparable to recent state-of-the-art (SoTA) while maintaining access to a low-dimensional latent code representation.

**Broader Impact Concerns:**

The current description of *Broader Impact Statement* is good.

**Requested Changes:**

Minimum adjustment:

- Please revise the paper based on the weakness ``Lack of comparisons & discussion with existing papers''.

Good-to-have adjustment:

- Besides the obvious better trade-off as presented at its form, is there any other novel insight the authors can deliver? For example, (1) the impact of the latent space for diffusion process; (2) the diffusion process for better learning of a meaningful VAE latent space, any chance to improve the latent space collapse issue [1,2]? Does learned representation yield better discriminative power?

[1] Avoiding Latent Variable Collapse with Generative Skip Models

[2] Cyclical Annealing Schedule: A Simple Approach to Mitigating KL Vanishing

**Strengths And Weaknesses:**

**Strengths**
- The paper is well written. The main idea is clearly presented, in terms of both method description and experiment comparison.
- To the best of my knowledge, DiffuseVAE is the first model to combine the two techniques: VAE and Diffusion models. Though the proposed idea is straightforward, but it comes a good motivation: Better speed vs quality tradeoff by combining the strengths of both models. This could be a useful data point to enrich the literature.
- The experiments are conducted to support the above strengths to combine the two models: High image generation quality, faster diffusion process compared with baseline DDPM, and controllable generation with a latent code.


**Weaknesses**
- Novel insight is lacking. I am afraid that the readers who are familiar with the literature cannot learn much new after reading the paper.
- Experiments are not exciting, though standard. The experiments are conducted are established datasets in the academy setting. The results are compared with some related models in terms of quantitative metrics such as FID & IS. The performance of DiffuseVAE is comparable to SoTA.
- Lack of comparisons & discussion with existing papers. There are two types of works can be discussed and compared in the paper:

  1. **VQVAE + Diffusion**. This paper combines standard VAE with diffusion models. It is straightforward to combine VQ-VAE with diffusion models, and perform the so-called two-stage training. In the literature, I'm afraid that similar ideas have been explored. For example, VQ-VAE is used to get the compact discrete representation of images. The diffusion process is applied in either the latent space or the pixel space. I strongly feel the proposed DiffuseVAE is closely related to this line of research, and should be thoroughly discussed and compared. Importantly,  VQVAE + Diffusion have been conducted in a larger scale and show more impressive empirical results and applications.
  2. **More papers that report good results on these standard datasets**, eg CIFAR-10, CelebA-64. There are several generative models that achieve better numbers, especially GANs [1,2]. For example, In Table 11(b) of [1], the proposed ADA techqniues yields FID 2.42 and IS 10.14. Please consider include all methods and numbers in Table 11(b) of [1] in Table 4 of this paper. The current Table 4 is misleading in this sense:  GANs are not as bad as it is currently presented.


[1] Training Generative Adversarial Networks with Limited Data, NeurIPS 2020

[2] Feature Quantization Improves GAN Training, ICML 2020

---

> ### Author Response · Authors · 2022-09-21
> **Response to Reviewer TFVU**
>
> We would like to thank the reviewer for their feedback on our work. We address specific concerns stated in the Weaknesses section as follows:
>
>
>
> 1.
> **Novel insight is lacking:**
> We would like to highlight the following novel methodological and empirical contributions of our work:
>
> **Methodological Contributions**
>
>
>
> * In this work, we present a novel framework which enables us to formally reason about learning the structure of conditioning signals and exploiting it in downstream generative models like Diffusion models. A special case of this framework is DiffuseVAE in which reconstructions from a simple VAE are refined using a powerful Diffusion model.
> * We also present a novel conditioning framework for diffusion models which allows conditioning both the forward and the reverse processes on the conditioning signal (which we call Formulation-2). This results in an updated training and sampling procedure (See Appendix B). In contrast, prior works [1,2] only condition the reverse process on the conditioning signal while assuming a condition independent forward process (denoted as Formulation-1 in this work). Interestingly, we observe that Formulation-2 provides a better speed vs quality tradeoff than Formulation-1 (For instance, DiffuseVAE Formulation-2 achieves a FID of **16.57 vs 18.01** achieved by Formulation-1 on the CelebA-HQ-128 benchmark when T=10 during DDIM sampling. See Table 3 for more details).
>
> **Empirical Contributions**
>
>
>
> * Our work is amongst the first works (See [3] for a concurrent approach) to equip diffusion models with a low-dimensional latent code, thus leveraging the advantages of both model families in a single model. We establish this empirically as presented in Sections 4.1, 4.2 and 4.4
> * Secondly, our work demonstrates a superior speed vs quality tradeoff during inference as compared to a baseline unconditional DDIM/DDPM (atleast **2x** and upto **10x** speedups on some benchmarks. See Table 3 and Section 4.3 for details). To the best of our knowledge, the impact of using a conditional signal on the speed vs quality tradeoff of diffusion models is relatively underexplored in the existing literature. Our work provides novel insights into this aspect by providing extensive comparisons with unconditional DDIM/DDPM on multiple benchmarks at multiple image resolutions.
>
>
>
> 2.
> **Choice of Experiments:** Our experiment design was motivated from the potential methodological advantages that we felt our model design has over both VAEs and Diffusion model families considered in isolation (like controllable synthesis, better speed-quality tradeoffs, higher fidelity samples etc). Our choice of training benchmarks was primarily influenced by limited compute capacity at our disposal and availability of evaluation metrics for prior methods. In this regard, the choice of experiments might seem standard but we feel this choice readily validates our claims and therefore justified. Moreover, we believe that our experimental results which show DiffuseVAE generalization to different noise types in the conditioning signals (See Sec. 4.5) might be interesting for the community as it motivates the design of conditional models which can generalize to unseen conditioning noise in a zero-shot fashion.
>
>
> 3.
> **Lack of comparisons with existing methods:** We thank the reviewer for pointing this out. We have updated Table 4 in our revision to include results from [4] for the unconditional StyleGAN2-ADA and related models.
>
> 4.
> **Discussion with existing papers**: We agree that a VQ-VAE model can also be used within the DiffuseVAE model framework. Though an interesting direction for future research, in this work, we work with the vanilla VAE model with standard Gaussian prior. In addition to the original work on VQVAE and VQVAE-2, other recent approaches which combine VQ-VAEs with other generative model families (like GANs, diffusion models) include VQGANs [5] and Latent Diffusion (LDM) [6]. LDM applies diffusion in the latent space of a pretrained VQGAN which is already a powerful autoencoding baseline. In contrast, our method refines “blurry” reconstructions generated by an extremely lightweight VAE using a downstream diffusion model. A possible benefit of having a generator-refiner framework in contrast to the LDM framework could be the requirement of a powerful VAE baseline as a pre-requisite in LDM in order to generate high-quality samples. Since there exists a trade-off between latent code disentanglement and high quality reconstructions [7], the need of a high fidelity autoencoding baseline can be disadvantageous in situations where a fine-grained control over the generated samples is required. We hypothesize that this problem is alleviated in DiffuseVAE since our first stage model can readily tradeoff more disentanglement for lower fidelity reconstructions due to a powerful second stage diffusion-based refiner model. We have included this discussion in Section 5 in our revision.

---

> > ### Author Response · Authors · 2022-09-21
> > **(Contd.)**
> >
> > 5.
> > **Good-to-have adjustments:**
> >
> >
> > * Impact of the z latent space on the diffusion process: We cover this aspect extensively in Section 4.2 of the paper.
> > * The diffusion process for better VAE latent spaces: In this work, since one of our primary goals is high quality image synthesis, we train DiffuseVAE in a two-stage fashion as opposed to end-to-end training (See Section 3.2, Appendix E). However, due to a 2-stage approach, the diffusion model training is separate from a VAE, and due to this the VAE latent codes cannot be improved. However, in an end-to-end DiffuseVAE setting, it might be interesting to explore the impact of joint training on the VAE latent space. We leave this to future work and have mentioned this point in Section 6 of our paper.
> >
> > **References:**
> >
> >
> > [1] Image Super-Resolution via Iterative Refinement by Saharia et al.
> >
> > [2] Cascaded Diffusion Models for High Fidelity Image Generation by Ho et al.
> >
> > [3]** **Diffusion Autoencoders: Toward a Meaningful and Decodable Representation
> >
> > [4] Training Generative Adversarial Networks with Limited Data by Karras et al.
> >
> > [5] Taming Transformers for High-Resolution Image Synthesis by Esser et al.
> >
> > [6] High-Resolution Image Synthesis with Latent Diffusion Models by Rombach et al.
> >
> > [7] beta-VAE: Learning Basic Visual Concepts with a Constrained Variational Framework

---

> ### Comment · Action_Editors · 2022-10-21
> **Please provide your recommendation**
>
> Dear reviewer,
>
> Please provide your recommendation for the paper.
>
> Best!

---

### Author Response · Authors · 2022-11-29
**Camera Ready version**

Dear Reviewers and Action Editor,

We have uploaded the camera-ready version of our paper and have added the link to our GitHub repo (with checkpoints). Compared to the last revision, we have removed the red color code which we used to denote changes between the initial and the revised manuscript. We would like to thank you for the constructive reviews and for acknowledging our contributions.

Best Regards,

Paper346 Authors

---

### Decision · Action_Editors · 2022-11-07

**Recommendation:** Accept as is

**Comment:**

The authors proposed a new combination of a VAE and a Diffusion-based model. It is the first (or at least one of the first) approaches to combine these two techniques. The paper is well-written and easy to follow. The results are mostly convincing and support the initial claims. After discussing the decision with the reviewers, I recommend to accept the paper as it is.

**Audience:**

The proposed research line is up-to-date with a potential great interest to the audience of TMLR. The combination of VAEs and Diffusion-based Moldes is very interesting.

**Claims And Evidence:**

The authors list out their claims in a ludic manner. Moreover, they provide enough empirical support for them. The reviewers rather agree with the statements of the authors (one reviewer would prefer to see some additional results, however, it is not a major and necessary addition).